# Joint inference of evolutionary transitions to self-fertilization and demographic history using whole-genome sequences

**Stefan Strütt[1†], Thibaut Sellinger[2,3†], Sylvain Glémin[4,5], Aurélien Tellier[2]\*, Stefan Laurent[1]\***

[1]Max Planck Institute for Plant Breeding Research, Cologne, Germany; [2]Department of Life Science Systems, Technical University of Munich, Munich, Germany; [3]Department of Environment and Biodiversity, Paris Lodron University of Salzburg, Salzburg, Austria; [4]Université Rennes 1, CNRS, ECOBIO, Rennes, France; [5]Department of Ecology and Genetics, Evolutionary Biology Centre, Uppsala University, Uppsala, Sweden

**\*For correspondence:**
aurelien.tellier@tum.de (AT);
laurent@mpipz.mpg.de (SL)

[†]These authors contributed equally to this work

**Competing interest:** The authors declare that no competing interests exist.

**Abstract** The evolution from outcrossing to selfing occurred recently across the eukaryote tree of life in plants, animals, fungi, and algae. Despite short-term advantages, selfing is hypothetically an evolutionary dead-end reproductive strategy. The tippy distribution on phylogenies suggests that most selfing species are of recent origin. However, dating such transitions is challenging yet central for testing this hypothesis. We build on previous theories to disentangle the differential effect of past changes in selfing rate or from that of population size on recombination probability along the genome. This allowed us to develop two methods using full-genome polymorphisms to (1) test if a transition from outcrossing to selfing occurred and (2) infer its age. The *teSMC* and *tsABC* methods use a transition matrix summarizing the distribution of times to the most recent common ancestor along the genome to estimate changes in the ratio of population recombination and mutation rates overtime. First, we demonstrate that our methods distinguish between past changes in selfing rate and demographic history. Second, we assess the accuracy of our methods to infer transitions to selfing approximately up to 2.5$N_e$ generations ago. Third, we demonstrate that our estimates are robust to the presence of purifying selection. Finally, as a proof of principle, we apply both methods to three *Arabidopsis thaliana* populations, revealing a transition to selfing approximately 600,000 years ago. Our methods pave the way for studying recent transitions to self-fertilization and better accounting for variation in mating systems in demographic inferences.

## Editor's evaluation

This manuscript details an important development of population genetics theory that can be used to infer past changes in the selfing rate of natural populations. The inference procedure is convincing and represents a substantial improvement upon previous methods. The work will be of broad interest to researchers studying mating system evolution and its consequences and will improve demographic inferences drawn from population genetic approaches.

## Introduction

Hermaphroditism is common in many groups of eukaryotes, especially in plants, and allows uniparental reproduction through selfing. The rate of self-fertilization is known to vary widely between species and populations; from exclusive outcrossing, through mixed mating, to predominant self-fertilization

(*Stebbins, 1950*; *Whitehead et al., 2018*). Transition from outcrossing to predominant self-fertilization is the most frequent reproductive transition in flowering plants (*Barrett, 2010*) and is thought to have occurred many hundreds of times. Such a transition has profound ecological and evolutionary consequences affecting the genetic functioning and the demography of a population as well as patterns of dispersal. Taking selfing rate and its possible variations into account is thus key to properly understand the evolution of fate of a species.

In flowering plants, cross-fertilization is ensured by diverse molecular and morphological features, some of which being referred to as self-incompatibility (SI) mechanisms (*Charlesworth, 2010*), which are defined as 'the inability of a hermaphroditic fertile seed plant to produce zygotes after self-pollination' (*de Nettancourt, 1997*). SI mechanisms enable the pistil of a plant to identify and repel self-pollen or pollen of a related genetic type and as a consequence, avoid inbreeding (*de Nettancourt, 1997*). SI systems are known to be encoded by a small number of genes and can therefore easily be lost. Indeed, genetic disruptions of SI systems through naturally occurring mutations are thought to be a major driver of plant reproductive diversity and have been linked to recent transitions to predominant self-fertilization in several species (*Shimizu and Tsuchimatsu, 2015*; *Mattila et al., 2020*).

On the short term, whether a new mutation responsible for an SI breakdown will invade the population or be lost depends on the balance between the main advantages of selfing (reproductive assurance and gene transmission advantage) and inbreeding depression, that is the reduced fitness caused by an increased homozygosity under inbreeding (*Charlesworth and Charlesworth, 1987*; *Charlesworth, 2006*). On the long term, selfing is predicted to increase extinction rate and to reduce diversification of selfing lineages as it has been observed in a few clades such as Solanaceae (*Zenil-Ferguson et al., 2019*), Primulaceae (*de Vos et al., 2014*); but see *Landis et al., 2018*. A consequence is the supposed recent origin of selfing species due to an excess of transitions on terminal branches. It thus nicely illustrates how the balance between micro-evolutionary and macro-evolutionary processes can generate the observed distribution of mating systems among species (*Igic et al., 2008*; *Goldberg et al., 2010*). This peculiar dynamic is also invoked to explain that the genomic effects of selfing are often detected on within-species polymorphism but very rarely on between-species divergence (*Glémin et al., 2019*). The recent timing of these transitions is thus a central assumption but has not been systematically tested as it remains a challenging task.

Despite the increasing number of available methods to reconstruct the evolutionary history of populations using genomic data (*Excoffier et al., 2013*; *Schiffels and Durbin, 2014*; *Boitard et al., 2016*; *Terhorst et al., 2017*; *Speidel et al., 2019*), none of them is currently able to explicitly identify and estimate the age of transitions in reproductive strategies. In flowering plants, previous attempts to estimate the age of transitions to selfing were based on limited genetic variation at a single locus directly controlling the reproductive mode in plants: the *S-locus* (reviewed in *Mattila et al., 2020*). For example, naturally occurring loss-of-function alleles at the *S-locus* were shown to be responsible for the loss of SI in *Arabidopsis thaliana* (*Tsuchimatsu et al., 2010*). The steady accumulation of non-synonymous alleles following loss of constraint at the *S-locus* was used to estimate that the age of the transition is at most 1.48 million years old, based on current estimates of the mutation rate (*Shimizu and Tsuchimatsu, 2015*). Note that the original upper-bound estimate in *Bechsgaard et al., 2006*, is 413,000 years owing to the use of a different mutation rate (see Figure 4 in *Shimizu and Tsuchimatsu, 2015*). This approach is limited by the small number of genetic variants upon which the estimation is conducted and can only be used in species for which, as is the case in *A. thaliana*, the genetic determinism of the loss of SI is known. However, a shift in reproductive system is expected to strongly impact genome-wide polymorphisms patterns, that is not only at the (*S-*)loci controlling it, thereby leaving a potentially characteristic molecular signature. We use this rationale and build upon previous theoretical work to develop two inference tools allowing to use full-genome polymorphism data of any species in order to (1) reveal the occurrence of past changes in reproduction mode and (2) estimate their age.

A classic way to consider selfing assumes a theoretical population of $N$ diploid individuals which produce offspring through selfing or outcrossing with probability $\sigma$ and $1 - \sigma$, respectively. Under a model of neutral evolution, the distribution of polymorphic sites in a sample of sequenced individuals, that is the frequency of alleles (single nucleotide polymorphism [SNPs]), is determined by the underlying genealogy of this population. A genealogy has for properties its length measured as the

time to the most recent common ancestor ($T_{\text{MRCA}}$) and its topology, that is the order and number of branching processes. Along the genome, genealogies change due to the effect of recombination (the so-called ancestral recombination graph [ARG]; *Hudson and Kaplan, 1988*; *Wiuf and Hein, 1999*). Two population parameters determine the distribution and characteristics of genealogies observed in a sample of several genomes: the population mutation rate ($\theta$) and the population recombination rate ($\rho$). In the presence of predominant selfing, the effective population size ($N_\sigma$) of a population of $N$ individuals is scaled by the selfing rate ($\sigma$) yielding $N_\sigma=N/(1+F)$ (*Fu, 1997*; *Nordborg and Donnelly, 1997*) and the recombination rate ($r_\sigma$) is scaled as $r_\sigma=r(1-F)$ (*Golding and Strobeck, 1980*; *Nordborg, 2000*; *Padhukasahasram et al., 2008*) at least when $r$ is not too high (*Padhukasahasram et al., 2008*), where $r$ is the recombination rate (for example per site) in the genome and $F$ is the inbreeding coefficient. When inbreeding is only due to partial selfing, $F=\sigma/(2-\sigma)$ and takes values between 0 and 1 (0 for outcrossing and 1 for fully selfing). As a consequence, the population recombination rate takes now for value in a selfing population:

$$\rho_\sigma = 4N_\sigma r_\sigma = 4Nr(1-F)/(1+F) \tag{1}$$

With $\mu$ being the mutation rate in the genome (e.g. per site), the population mutation parameter accounts for the effect of selfing as follows:

$$\theta_\sigma = 4N_\sigma\mu = 4N\mu/(1+F) \tag{2}$$

We note that the classic ratio of population recombination by population mutation rate $\rho/\theta=r/\mu$ in outcrossing species becomes now with selfing $\rho_\sigma/\theta_\sigma=r(1-F)/\mu$ (*Nordborg and Donnelly, 1997*; *Möhle, 1998*; *Nordborg, 2000*). Taken together, expressions (*Equations 1; 2*) suggest that selfing does not amount to a simple change in effective population size (from $N$ to $N_\sigma$), and that the reduction of the effective recombination rate is more severe than the reduction in effective population size.

Following these insights, two key predictions can be derived. First, a characteristic and specific signal of selfing, in contrast to outcrossing, is expected to be present in SNP data due to the joint action of recombination along the genome (rate $\rho_\sigma$) and of the genealogical (coalescence) process (rate $\theta_\sigma$). The first prediction underlies the previous development of a sequentially Markov coalescent method (eSMC) to estimate a fixed selfing rate using estimations of the ratio $\rho_\sigma/\theta_\sigma$ from pairs of genomes (*Sellinger et al., 2020*). Second, evolutionary changes in reproductive systems (transition from selfing to outcrossing or vice versa) are reflected in variations of the ratio $\rho_\sigma/\theta_\sigma$ in time and are identifiable and distinguishable from changes in population sizes alone. The latter suggests that a characteristic signature of the change in selfing rate in time should be observed in polymorphism data along the genome, if genetic variation can be summarized in a way that is informative about the joint effect of genetic drift and recombination. Indeed, it is desirable to (1) estimate changes in population size which occur independently of a transition, for example when species colonize new habitats as facilitated by selfing, and (2) disentangle the possible confounding effect of population-size changes on the estimation and detection of a transition. To our knowledge no statistical inference method exists that takes advantage of these theoretical predictions to jointly estimate temporal changes in selfing rates and in population sizes.

Two types of model-based methods are used to draw inference of past demographic events using full-genome polymorphism data. First, the distributions of the time to the most recent common ancestor ($T_{\text{MRCA}}$) along a pair of chromosomes sampled from a sexually reproducing population can be modeled and approximated assuming the sequentially Markov coalescent (SMC) (*McVean and Cardin, 2005*). SMC-based methods infer model parameters, such as demographic histories, from fully sequenced genomes and have been implemented in several statistical software used to estimate changes in effective population size through time under the assumption of a Wright-Fisher (WF) model. *Li and Durbin, 2011*, developed a pairwise SMC estimating the distribution of $T_{\text{MRCA}}$ over two sequences, and *Schiffels and Durbin, 2014*, extended the framework to consider multiple (more than two) sequences at a time, albeit only estimating the most recent pairwise $T_{\text{MRCA}}$. These SMC-based methods are based on a hidden Markov model (HMM) and the forward-backward algorithm for estimating the ARG (i.e. the genealogies along the sequence) and a Baum-Welch algorithm to estimate the parameters of the model. In our previously developed SMC method, *eSMC*, we included the effect of constant-in-time seed banks and self-fertilization in order to estimate these parameters (i.e. dormancy and selfing rates) jointly with past population sizes (*Sellinger et al., 2020*). eSMC uses

also pairs of sequences and the estimated transition matrix which summarizes the transition between two consecutive genealogies along the genomes (due to a recombination event), in contrast to the computationally intensive approach based on the actual series of coalescence times (*Gattepaille et al., 2016*). The SMC methods are characterized by their easy applicability to datasets as these do not require the user to specify a given demographic model. However, the SMC methods rely on an analytical expression for the transition probability between genealogies and do not provide a measure of uncertainty of the parameters or allow for hypothesis testing between different models/scenarios (though in principle the transition matrix can be used to do so, *Palacios et al., 2019*). Second, approximate Bayesian computation (ABC) is a computational approach to estimate posterior probabilities for models and parameters that is well suited for demographic modeling with many parameters and without any analytically derived likelihood function (*Beaumont et al., 2002*; *Csilléry et al., 2010*). Two advantages of the ABC method are that it allows to compare competing demographic hypotheses on the basis of Bayes factors and it does not require bootstrapping the data to generate measures of uncertainty for the inferred parameters. A critical aspect of ABC is that it requires a careful summarization of the genomic data into a set of summary statistics that carry information about the parameters of interest (*Beaumont et al., 2002*). This step is specially challenging when using genome-wide data (*Boitard et al., 2016*). With the aim to use the information on genetic drift and recombination present in full-genome polymorphism data, we develop therefore both an SMC and an ABC approach to infer past demographic events and the time of transition to selfing. We first provide analytical and simulation results explicating the consequences of a transition to selfing on genomic variation, thereby confirming our second prediction that temporal changes in selfing rate leave observable specific patterns in polymorphism data across the genome. Then, we analyze the statistical accuracy of the two newly developed methods to identify and estimate the age of transitions to selfing using a small number of sequenced genomes. Third, as a proof of principle, we apply these methods to estimate the age of the transition to selfing in *A. thaliana*, in which it has been documented and for which full-genome polymorphism data exist. These new methods are useful toolkits for dating and understanding the evolution of the selfing syndrome and shifts in breeding systems and reproduction modes.

## Results

### The consequences of a transition to selfing on patterns of genomic variation

We consider a theoretical population of $N$ diploid individuals (equivalent to $2N$ chromosomes), which produce offspring through selfing or outcrossing with probability $\sigma$ and $1 - \sigma$, respectively (following the notations in *Nordborg, 2000*). At some time ($t_\sigma$) in the past, the previously outcrossing population (with $\sigma=0$) undergoes a transition to selfing and remains selfing until present (with a selfing rate $\sigma>0$). Independently of the change in selfing rate, the population size can change once from $N_{\text{ANC}}$ (ancestral) to $N_{\text{PRES}}$ (present) at time $t_N$ (measured from the present). We implement the selfing model both in the forward WF framework, in which selfing can be simulated explicitly, and using the coalescent-with-selfing, in which selfing is modeled through a rescaling of the effective population size and of the recombination rate at $t_\sigma$ (*Nordborg and Donnelly, 1997*; *Möhle, 1998*; *Nordborg, 2000*) (see Materials and methods). In other words, selfing rate is not constant in time in our model.

In the spirit of the SMC approaches, we obtain for our model a first analytical result extending the previous theoretical work to consider the transition between two consecutive genealogies along the genome due to a recombination event (here reduced to the coalescence time of a sample of size two). In contrast to the classic assumption of SMC methods (*Li and Durbin, 2011*; *Schiffels and Durbin, 2014*; *Palacios et al., 2019*) the ages of recombination events do not follow here a uniform distribution along the genealogy, but rather are functions of the selfing and recombination rates at each time point. As recombination and selfing rates are assumed non-constant through time, the probability for a recombination event to occur follows an inhomogeneous Poisson process along the coalescent tree. We thus compute the probability for an effective recombination event to occur in a sample of size two ($p(rec|s)$), conditioned to the current coalescence time $s$, and with the selfing and recombination rates at time $k$ being, respectively, $\sigma_k$ and $r_k$. We find (using *Equation 1* for recombination above):

$$\mathrm{p}\left(\mathrm{recls}\right) = \left(1 - \mathrm{e}^{-\int_0^s \frac{2(1-\sigma_k)}{(2-\sigma_k)} 2\mathrm{r}_k \mathrm{dk}}\right) \tag{3}$$

Note that this value depends in an inhomogeneous way on the current coalescent time, which requires to integrate along the branch length of the coalescent tree. In other words, the variation of population size through time does not affect the probability of a recombination to occur conditioned on the coalescence time, that is the variation of population size affects the coalescence rate only. In *Equation 3* it is visible that the probability for a recombination event to occur and to modify the genealogy is affected by the selfing rate ($\sigma_k$). Thus, this result opens the possibility to jointly infer piecewise functions of selfing or recombination rates and population sizes through time simultaneously. In practice, a change in selfing rate can be detected when the span of the genealogy along the genome does not decrease homogeneously with increasing coalescence time as demonstrated in *Figure 1—figure supplement 1* (based on *Equation 3*). A further formal description of the coalescence process with variable recombination and/or selfing rates through time in the context of the SMC is provided in Appendix 1, especially providing the required expressions for the transition and emission probabilities needed to compute the transition matrix.

Following the spirit of SMC methods, we now turn to simulations to explicit the importance of *Equation 3* for inference. We thereafter verify the second prediction - temporal changes in selfing rate leave observable specific patterns in polymorphism data across the genome - by quantifying the expected distributions of $T_{\mathrm{MRCA}}$-segments (hereafter TL-distribution) under our model. These segments are defined as successive and adjacent sets of nucleotidic positions sharing the same time to the most recent common ancestor ($T_{\mathrm{MRCA}}$) and are separated by the breakpoints of ancestral recombination events (*McVean and Cardin, 2005*). Segments are summarized by their lengths and $T_{\mathrm{MRCA}}$ (*Figure 1A and B*). When the selfing rate is constant the TL-distribution has a negative constant covariance, because, on average, older segments are exposed, and shortened by a larger number of recombination events compared to younger segments (*Figure 1A*). In the case of a transition from outcrossing to predominant selfing, the rate at which the segments shorten with age is dramatically increased in the outcrossing phase compared to the selfing phase (*Figure 1B*), leading to a characteristic change in the covariance of the joint distribution at $t_\sigma$. This behavior of the model is invariably observed, using our coalescent model rescaling $\rho$ and $\theta$, or by explicitly using forward WF simulations (*Figure 1—figure supplement 2*). Importantly, this specific genomic signature of a transition to selfing is not observed when the selfing rate is constant and only the population-size changes from $N_\sigma$ to $N$, where $N_\sigma$ would be the effective population size of a population with selfing rate $\sigma$ (*Figure 1A*). The simulations confirm thus the results from *Equation 3* and *Figure 1—figure supplement 1*.

Although the same increase in $\rho$ could in principle be accounted for by a very large ancestral population size (i.e. $N_{\mathrm{ANC}} = N_{\mathrm{PRES}}/(1 - F)$), such a model would also have a much larger ancestral $\theta$ compared to the selfing model, and thus cannot produce the same TL-distribution as a transition to selfing (and thus is not a confounding scenario for a transition to predominant self-fertilization). The TL-distribution is determined by the probability of recombination events, which, as explicated in *Equation 3*, is affected by the change in selfing ($\sigma$) when conditioning on a given $\theta$. Simulated TL-distributions under a range of values for $t_\sigma$ illustrate the dependency of the change in the covariance between the age and length of $T_{\mathrm{MRCA}}$-segments on the age of the transition (*Figure 1—figure supplement 3*). Thus, this suggests that genome-wide polymorphism data contains information about shifts to selfing when the age of the transition falls well within the distribution of $T_{\mathrm{MRCA}}$.

We also made the important observation that all the segments that coalesce in the outcrossing phase ($T_{\mathrm{MRCA}}$ older than $t_\sigma$) are spatially clustered along simulated chromosomes (*Figure 1C*). This effect can also be captured by inspecting the occurrences of transitions between different $T_{\mathrm{MRCA}}$ for a large number of successive and adjacent fragments (*Figure 1D and E*). In the case of a transition to selfing, the $T_{\mathrm{MRCA}}$ transition matrix distinctly shows that segments with $T_{\mathrm{MRCA}}$ older than $t_\sigma$ are more likely to be followed by segments that are also older than this time. Although a similar dependency between successive $T_{\mathrm{MRCA}}$ also exists when selfing is constant, the magnitude of the effect is more pronounced in the case of a shift to selfing. The specific genomic signatures of a transition to selfing and its equivalence to two simultaneous temporal changes in effective recombination rate and population size motivated us to design two new statistical methods to estimate the age of transitions to selfing.

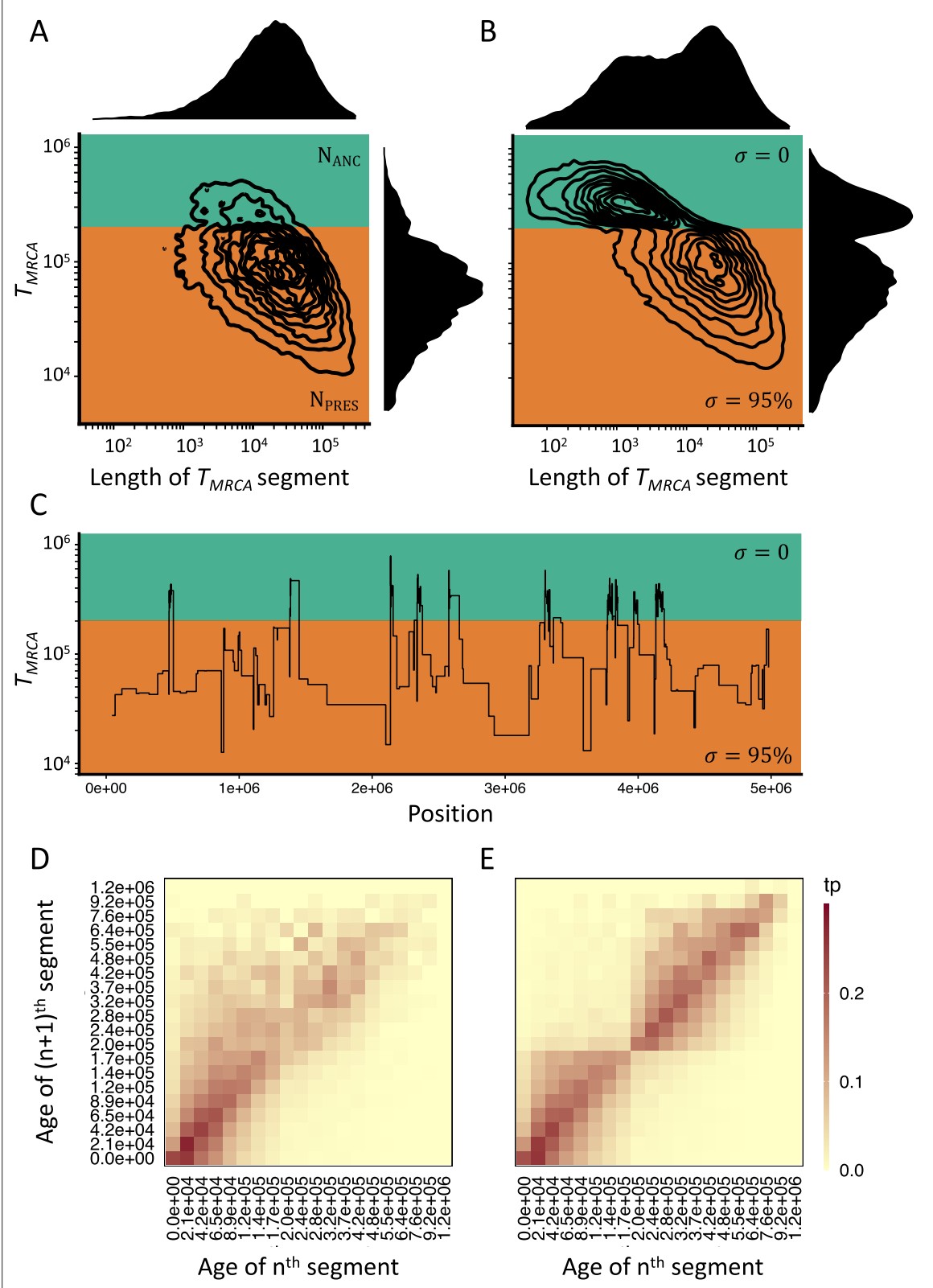

**Figure 1.** Consequences of a transition to selfing on the genealogies of simulated chromosomes. (**A**) Joint and marginal distributions of ages ($T_{MRCA}$ in generations on a log10 scale) and lengths of $T_{MRCA}$-segments (in bp on a log10 scale) in a selfing population ($\sigma$=0.95) with a stepwise change from large (green, $N_{ANC}$ = 50,000) to low (orange, $N_{PRES}$ = 26,250) population size. The population sizes were chosen to correspond to the rescaling of the effective population size by the selfing rates used in panel B. (**B**) Joint and marginal distributions of ages ($T_{MRCA}$) and lengths of $T_{MRCA}$-segments (in bp) in

*Figure 1 continued on next page*

*Figure 1 continued*

a population with a constant population size and a shift from outcrossing (green, $\sigma$=0) to predominant selfing (orange, $\sigma$=0.95). (**C**) Spatial distribution along the genome of a subset of the $T_{MRCA}$-segments simulated in panel B ($t_\sigma$=200,000 generations). (**D**) The transition matrix of ages ($T_{MRCA}$) between adjacent segments along the genome for the data simulated in panel A. This matrix summarizes the probabilities that the $n$th $T_{MRCA}$-segment with a given age $X$ is followed by the ($n$+1)th segment of age $Y$. The heat colors indicate the transition probabilities (tp). (**E**) The transition matrix of ages ($T_{MRCA}$) between adjacent segments along the genome for the data simulated in panel B. The recombination rate for the simulations was set to 3.6×10⁻⁹. The data was acquired by conducting 20 independent replications (see Materials and methods).

The online version of this article includes the following figure supplement(s) for figure 1:

**Figure supplement 1.** Probability of recombination events over time under different reproductive histories.

**Figure supplement 2.** Comparison of the joint distributions of $T_{MRCA}$ and lengths of $T_{MRCA}$-segments (TL-distribution) under three different simulation approaches.

**Figure supplement 3.** Consequences of a transition to selfing on the genealogies of simulated chromosomes for different ages of the transition.

## Statistical methods to estimate the age of a transition to selfing: *teSMC*

Based on *Equation 3*, we extend *eSMC* into *teSMC*, allowing the estimation of varying selfing or recombination rates through time, jointly with varying population size (see Materials and methods and Appendix 1). In order to account for prior knowledge, two modes are implemented for parameter inference: (1) the free mode, in which each hidden state has its own independent selfing/recombination rate, and (2) the single-transition mode in which *teSMC* estimates only three parameters (the current and ancestral rates, and the transition time between both rates), a constraint greatly reducing the number of inferred parameters and well suited for the analysis of recent and sudden shifts from outcrossing to predominant self-fertilization.

First, to demonstrate the theoretical accuracy of our model and inference method, we analyze its performance when sequences of $T_{MRCA}$ are given as input (the age of coalescent trees for two samples). This is termed the best-case convergence of *teSMC* (*Sellinger et al., 2021*). We simulate data from a population undergoing a strong bottleneck and simultaneously a transition to selfing or a change in recombination rate. In both cases the population size and the past selfing or recombination values are recovered with high accuracy (*Figure 2—figure supplement 1*). Second, to understand the convergence properties of *teSMC*, we analyze its performance under a simple scenario assuming a constant population size and a constant selfing value of 0.9 given different amount of data. We compare the *eSMC* method, which estimate a constant rate of selfing in time with *teSMC*, which estimates varying selfing through time (*Figure 2—figure supplement 2*). When selfing is known to be constant (*eSMC*), the value of this parameter is recovered with high accuracy and low variance even with the lowest amount of given data (*Figure 2—figure supplement 2A*). When each hidden state can have its own selfing value (*teSMC*), constant selfing rate is indeed correctly recovered but a greater amount of data is required to reduce the variance in the estimation (*Figure 2—figure supplement 2B*).

We now evaluate the statistical accuracy of *teSMC* on polymorphism data from genome pieces of 5 Mb, simulated under a model with constant population size (N=40,000) with mutation ($\mu$) and recombination ($r$) rates of 1×10⁻⁸, and with an instantaneous change from outcrossing ($\sigma_{ANC}$ = 0.1) to predominant selfing ($\sigma_{PRES}$ = 0.99) at time $t_\sigma$ (see Materials and methods). Both the single-transition mode and the free mode estimation procedures perform well over the complete range of $t_\sigma$ values (*Figure 2A*).

Population sizes estimated under the assumption of a constant selfing rate are consistently larger than the true value in the outcrossing phase and display large fluctuations in the selfing phase, which could be mistaken for past population-size bottlenecks (*Figure 2—figure supplement 3*). However, when *teSMC* is allowed to account for the change in selfing rate, population-size estimates (N) remain close to the true values. We note that the increased variance in N in the recent selfing phase are likely caused by a smaller number of available $T_{MRCA}$-segments.

Finally, we evaluate the ability of *teSMC* to jointly estimate the age of a transition to predominant selfing and the time of a stepwise change in population size. To do this we use simulated data produced as above, except with the addition of a single stepwise population-size reduction (*Figure 2B and C*) or expansion (*Figure 2D and E*). In both cases our results indicate that *teSMC*, especially the free mode inference method, is able to precisely estimate the age of the shift to selfing, regardless of

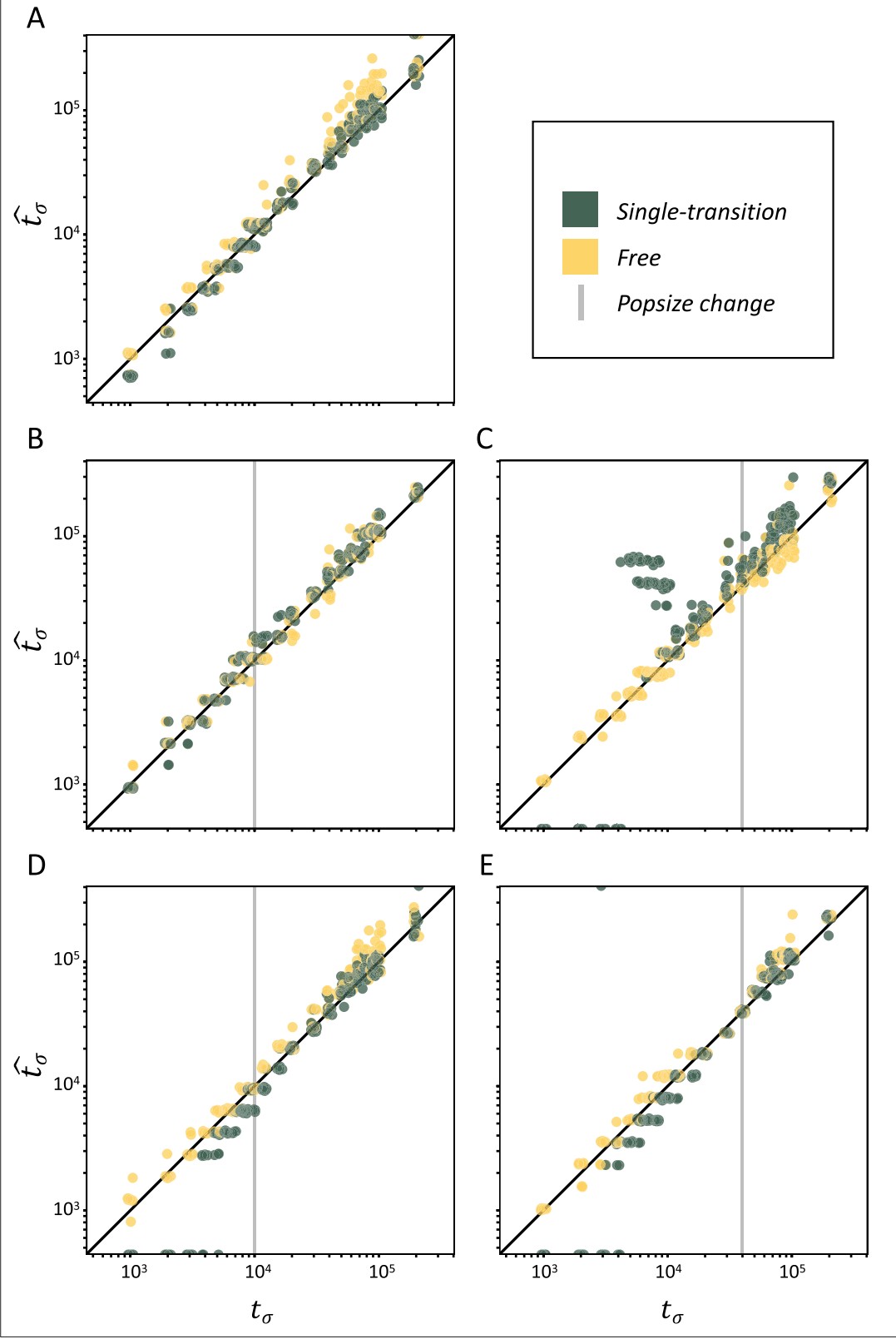

**Figure 2.** Performance of *teSMC* on simulated polymorphism data. Inference of times of transition from outcrossing ($\sigma$=0.1) to predominant selfing ($\sigma$=0.99) using neutral simulations. The x-axes represent the true value of $t_\sigma$ in units of log10(generations) and the y-axis shows the values of $t_\sigma$ estimated by *teSMC*. Inference was done using the free mode (yellow) and the one-transition mode (green) of *teSMC* and 10 replicates per time point.

*Figure 2 continued*

(**A**) Under constant population size. (**B–E**) Simulations were done with an additional change in population size, the vertical gray line indicates the age of the change in population size. (**B–C**) From $N_{ANC}$ = 200,000 to $N_{PRES}$ = 40,000 (population crash) at 10,000 generations (**B**) or 40,000 generations (**C**) in the past. (**D–E**) From $N_{ANC}$ = 40,000 to $N_{PRES}$ = 200,000 (population expansion) at 10,000 generations (**D**) or 40,000 generations (**E**) in the past. The inference process was repeated 10 times for each experimental condition, employing independently simulated data sets.

The online version of this article includes the following figure supplement(s) for figure 2:

**Figure supplement 1.** Theoretical convergence of *teSMC* under complex demography.

**Figure supplement 2.** Best-case convergence of *teSMC* for different amount of data.

**Figure supplement 3.** Mis-inference of population sizes when transitions to selfing are not accounted for.

**Figure supplement 4.** Inference of population sizes and selfing rates estimated by *teSMC* when both parameters change over time.

**Figure supplement 5.** Estimated population sizes by *teSMC* with variable mutation and recombination rates along the genome.

**Figure supplement 6.** Estimated selfing rates through time by *teSMC* with variable mutation and recombination rates along the genome.

**Figure supplement 7.** Best-case convergence of *teSMC* under complex selfing transitions.

**Figure supplement 8.** Performance of *teSMC* under complex selfing transitions.

---

the relative timing of the population-size change and the transition to selfing. Also, in most cases, the population sizes inferred by *teSMC* are close to the true simulated values (***Figure 2—figure supplement 4***). However, when the transition is recent and the present population size is low, the precision of the population-size estimates decreases (***Figure 2—figure supplement 4O***). We note that *teSMC* fails, in the latter case, to recover the population size, suggesting a lack of data (coalescent events) (***Sellinger et al., 2021***). These results demonstrate that transitions to predominant self-fertilization and more generally large changes in recombination rate through time can be captured by *teSMC* and the estimations can be disentangled from changes in population sizes.

## Statistical methods to estimate the age of a transition to selfing: *tsABC*

In the case of a transition to selfing, we require that the summary statistics used in the ABC are informative about coalescence and recombination rates, in order to make changes in selfing rates and population size distinguishable by the ABC model choice (***Figure 3***). Note that while the lengths of $T_{MRCA}$-segments are straightforward to calculate on simulated genealogies (***Figure 1A and B***), a main task is to estimate the length and age of these segments based on genomic diversity data. Following the results from *teSMC* using the highly informative transition matrix which summarizes the distribution of recombination events and of $T_{MRCA}$ for two successive coalescent trees along the genome (***Figure 2—figure supplement 1***), we derive new summary statistics for our ABC. Namely, we sum the number of nucleotide differences (i.e. SNPs) between pairs of sampled chromosomes using non-overlapping genomic windows of size $\omega$ (set to 10 kb throughout the whole study) and construct a transition matrix for pairwise diversity (displayed in ***Figure 1D and E***). This means that we evaluate the proportion of genomic windows with a given diversity value ($\pi$) transitioning to the adjacent window with another (or similar) value of diversity. In effect, this ABC transition matrix should tend to the input of teSMC when $\omega$ tends to 1. We refer to this summarization as $TM_{win}$ (***Figure 3—figure supplement 1***, see Materials and methods). The rationale behind this summarization is that, if the window size is smaller than some of the clusters of $T_{MRCA}$-segments in the outcrossing phase, then, $TM_{win}$ will capture both the expected excess of diversity of those segments and their clustering (***Figure 1C***). For comparison we also considered classic summarization based on the site frequency spectrum (SFS) and a discretized distribution of the decay in linkage disequilibrium (LD-decay), as these carry information about temporal changes in population size and selfing rates (***Tang et al., 2007***; ***Boitard et al., 2016***) based on the theory outlined in the introduction. We therefore evaluated the efficiency of three sets of summary statistics: SFS/LD, $TM_{win}$, and SFS/LD/$TM_{win}$ (see Materials and methods).

To test whether our approach can distinguish a transition to selfing from a reduction in population size, we conduct an ABC model choice analysis using two competing models (***Figure 3A and B***): model 1 where the selfing rate changes from $\sigma_{ANC}$ to $\sigma_{PRES}$ at $t_\sigma$ and with $N$ constant, and model 2

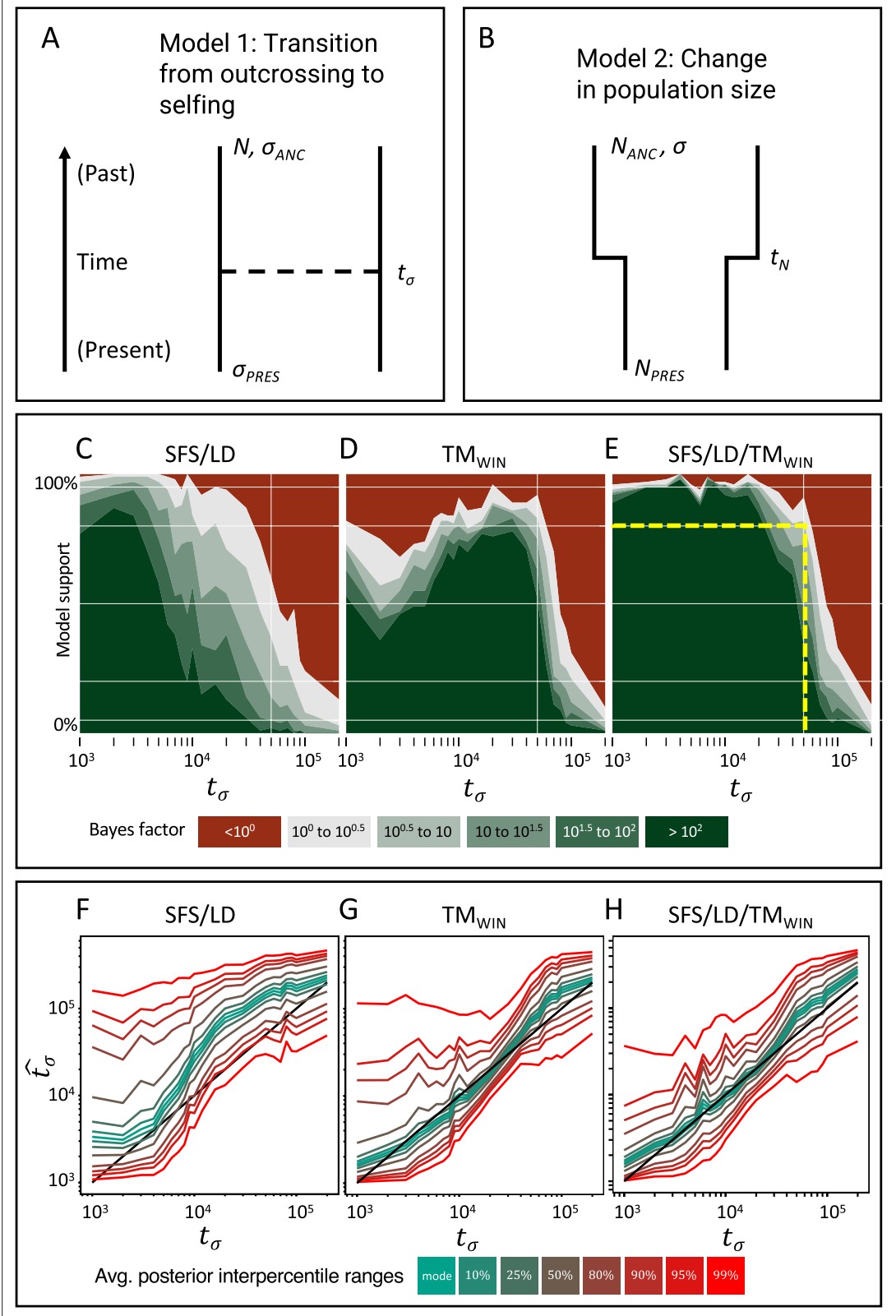

**Figure 3.** Approximate Bayesian computation (ABC) model choice performance analysis. (**A**) Demographic model 1 in the model choice analysis: one population with a single transition from predominant selfing to predominant outcrossing. (**B**) Demographic model 2 in the model choice analysis: one population with constant selfing and a single change in population size. The parameters of interest are the population sizes ($N_{PRES}$, $N_{ANC}$), the selfing rates ($\sigma_{ANC}$, $\sigma_{PRES}$), and the time of change in selfing rate and size ($t_\sigma$, $t_N$). (**C, D, E**) Performance of the ABC model choice method using three

*Figure 3 continued*

different summarizations of data and different values of $t_\sigma$ (x-axis). (**C**) Combining site frequency spectrum (SFS) and linkage disequilibrium (LD). (**D**) Window-based transition matrix (TM$_{win}$). (**E**) The combination out of SFS, LD, and TMwin. The x-axis represents the investigated range of $t_\sigma$ values in log10(generations). The y-axis indicates how often, for 100 datasets simulated under model 1, tsABC correctly identified the transition-to-selfing model. The yellow dashed lines indicate that, for a Bayes factor of approximately 3 (BF=√10), tsABC identifies the right model 80% of the time, for transitions as old as 51,000 generations (corresponding to 2.5$N_e$ generations in coalescent time units, where $N_\sigma$ is the effective population size of the selfing population). (**F, G, H**) Parameter estimation accuracy for the age of a transition to selfing (100 simulated datasets) under a model with constant population size ($N$=40,000) and a change in selfing rate from $\sigma_{ANC}$ = 0.1 to $\sigma_{PRES}$ = 0.99. Colored lines represent average quantiles for 100 posterior distributions.

The online version of this article includes the following figure supplement(s) for figure 3:

**Figure supplement 1.** Pairwise diversity transition matrices used in *tsABC*.

**Figure supplement 2.** Approximate Bayesian computation (ABC) performance analysis.

with population size changes from $N_{ANC}$ to $N_{PRES}$ and $\sigma$ constant. We simulate datasets under model 1 and evaluated the ability of *tsABC* to identify the correct model for transitions of varying ages, using different sets of summary statistics to summarize the genetic data (see Appendix 2). Our results show that our method recovers the correct model for transitions as old as 2.5$N_\sigma$ generations (*Figure 3C–E*), with $N_\sigma$=$N$/(1+$F$) being the effective population size for a selfing population. Interestingly, summarizing the genetic data using the SFS and LD-decay yields better performance for recent shifts to selfing (*Figure 3C and D*), while using TM$_{win}$ performs better for shifts occurring between (0.5$N_\sigma$ and 3$N_\sigma$); such that combining both set of summary statistics yields the best performance. This analysis confirms that shifts to selfing as old as approximatively 2.5$N_\sigma$ can be detected and can be disentangled from changes in population size using the ABC model choice procedure if appropriate summary statistics are used.

Next, we evaluate the accuracy of our method for estimating the age of a transition to selfing ($t_\sigma$). We simulate 100 datasets under model 1 (*Figure 3A*) with values of $t_\sigma$ ranging from 1000 to 200,000 generations, and used *tsABC* to re-estimate posterior distributions for $t_\sigma$ and the other parameters of the model (*Figure 3F–H*, *Figure 3—figure supplement 2*). Estimations are obtained using the same three summarization strategies used for the model choice (SFS/LD, TM$_{win}$, SFS/LD+TM$_{win}$). The age of a shift to selfing could be well estimated using the TM$_{win}$ approach, while the SFS+LD approach overestimated $t_\sigma$ almost over the complete range of values (*Figure 3F and G*). Combining SFS, LD, and TM$_{win}$ does not further improve the accuracy of the estimations (*Figure 3H*). We note that the parameters $N$ and $\sigma_{PRES}$ (i.e. the population size and the current selfing rate) are both better estimated with TM$_{win}$ than with SFS/LD, except for transitions younger than $10^4$ generations ago where $\sigma_{PRES}$ is slightly better estimated with SFS/LD (*Figure 3—figure supplement 2D*). However, no set of summary statistics could estimate the ancestral selfing rate (*Figure 3—figure supplement 2G–I*).

## Robustness of inference to model violations

We here provide two analyses to demonstrate the robustness of our inference method to two violations of the model assumptions: first recombination and mutation rates may vary along the genome, and second, background selection (BGS) in combination with selfing may affect the inference of transition to selfing. First, to assess the potential limits of our approach, we analyze the performance of *teSMC* when mutation and recombination rates are potentially non-constant along the genome (*Figure 2—figure supplements 5–6*). When mutation and recombination rates are constant along the genome, *teSMC* recovers a constant population size and accurate selfing rates (*Figure 2—figure supplements 5A and 6A*). When recombination rate varies by a twofold factor along the genome, the estimation of population size is accurate (*Figure 2—figure supplement 5B*) but the variance of selfing rates inference increases (*Figure 2—figure supplement 6B*). Variation of the mutation rate by a twofold factor along the genome biases inference through time, leading to erroneously high inferred selfing rates and population size (*Figure 2—figure supplement 2C, D*). Yet, these results are reassuring as they demonstrate that no spurious transition from outcrossing to selfing in the past is inferred under variation of the recombination rate or mutation rate along the genome.

BGS refers to the effect of deleterious alleles on linked neutral diversity (*Charlesworth et al., 1993*; *Irwin et al., 2016*). Recently, several studies highlighted that neglecting the effect of BGS in demographic analyses can lead to statistical biases and potential miss-identification of population-size

changes (*Ewing and Jensen, 2016*; *Johri et al., 2021*). Because transitions to selfing result in strong reduction of the effective population size and recombination rate (*Equations 1; 2*), a corresponding increase of linkage between deleterious and neutral alleles occurs (*Charlesworth et al., 1993*). Selfing indeed drastically magnifies the effect of BGS (*Roze, 2016*), so we test whether selfing and BGS could affect the accuracy of our inference methods. Because both the *teSMC* and the *tsABC* methods ignore the effect of selection, we evaluate their performance when applied to data simulated under a model with both a transition to selfing and BGS. We use *slim3* (*Haller and Messer, 2019*) to simulate genomic data with the same distribution of exonic sequences as in five pre-defined regions of 1 Mb from the genome of *A. thaliana* (exact coordinates in methods) and model negative selection on exonic sequences according to the distribution of fitness effects (DFEs) published by *Hämälä and Tiffin, 2020*. We found that when exonic sequences are not excluded from the analysis (unmasked), *teSMC* slightly underestimate the time of transition to selfing. However, when coding regions simulated with selection are masked the date of transition is accurately recovered (*Figure 4A*) and the performance is similar to the case without BGS (*Figure 2A*). We note that the *tsABC* estimations remain accurate even without masking exonic regions (*Figure 4B*, *Figure 4—figure supplement 1*) and the performance similar to the case without BGS (*Figure 3H*). These results suggest that our approach is generally robust to the effect of negative selection on linked neutral sites, even in compact genomes such as the one of *A. thaliana* (*Figure 4*).

## Application to *A. thaliana*

Deactivation of the SI mechanism through mutations knocking out the genes SCR and *SRK* is known to be responsible for the transition to predominant self-fertilization in the model species *A. thaliana* (*Tsuchimatsu et al., 2010*). This shift to selfing is the focus of several studies and estimates of its age have been obtained using different types of data and statistical approaches. Rescaling the original estimate with the more recent mutation rate of *Ossowski et al., 2010*, *Bechsgaard et al., 2006*, estimation of the oldest possible age of the transition is 1.48 million years (*Shimizu and Tsuchimatsu, 2015*); and was obtained using phylogenetic analyses of the *S-locus* in *A. thaliana* and *Arabidopsis lyrata*. *Tang et al., 2007*, analyzed the genome-wide decay in LD but could not detect the expected signature of a transition to selfing, and therefore concluded that the shift must have been older than the oldest coalescent events in their sample (older than 1 million years approximately). Here, estimations obtained with *teSMC* and *tsABC* range from 592,321 to 756,976 depending on which method and population samples are used (*Figure 5A*, *Table 1*, *Supplementary file 1*, Table S6). Our estimates are younger than the one proposed by *Bechsgaard et al., 2006*, and *Tang et al., 2007*. The probability that Bechsgaard's maximum age of the transition is sampled from the posterior distributions for $t_\sigma$ obtained with *tsABC* is equal to zero, as it is much older than the upper boundary of the corresponding credibility intervals (*Table 1*). We find the estimates of transition to selfing to be robust to the geographical origin of the population samples (Iberian non-relicts, Iberian relicts, or central European, *Figure 5*, *Supplementary file 1*, Tables S5 and S6). Note that we are also able to jointly estimate the demography of each *A. thaliana* population and the transition to predominant self-fertilization (*Figure 5B*). Genetic variants used for these estimations were taken from genomic intervals outside of pericentromeric regions, which are associated with above-average levels of nucleotide diversity in *A. thaliana* without elucidated reason yet (*Schmid et al., 2005*; *Clark et al., 2007*). Replicating this inference with a different set of loci resulted in very similar results (*Figure 5—figure supplement 2*). For application of *tsABC* and *teSMC* to other species, we therefore recommend to first investigate whether a similar association between pericentromeric regions and levels of diversity is observed and more generally whether some genomic regions exhibit peculiar patterns of diversity.

Compared to previous results ignoring the transition to selfing (*Sellinger et al., 2020*), we find a population-size decline starting at the time of transition and not continuously declining since the far past, as in previous studies (*Sellinger et al., 2020*). Our results suggest that population-size decline could be linked to the transition to selfing and that our demographic inference using *teSMC* should be more reliable than those from *eSMC* (and other SMC approach ignoring the selfing transition). A way forward would be to improve our inference framework using the new theoretical results by *Upadhya and Steinrücken, 2022*, which focuses on modeling the time to the most recent common ancestor of a sample size bigger than two. This study demonstrates that the inference in the far past can be improved, at the cost of losing accuracy in recent times. The estimated effective population

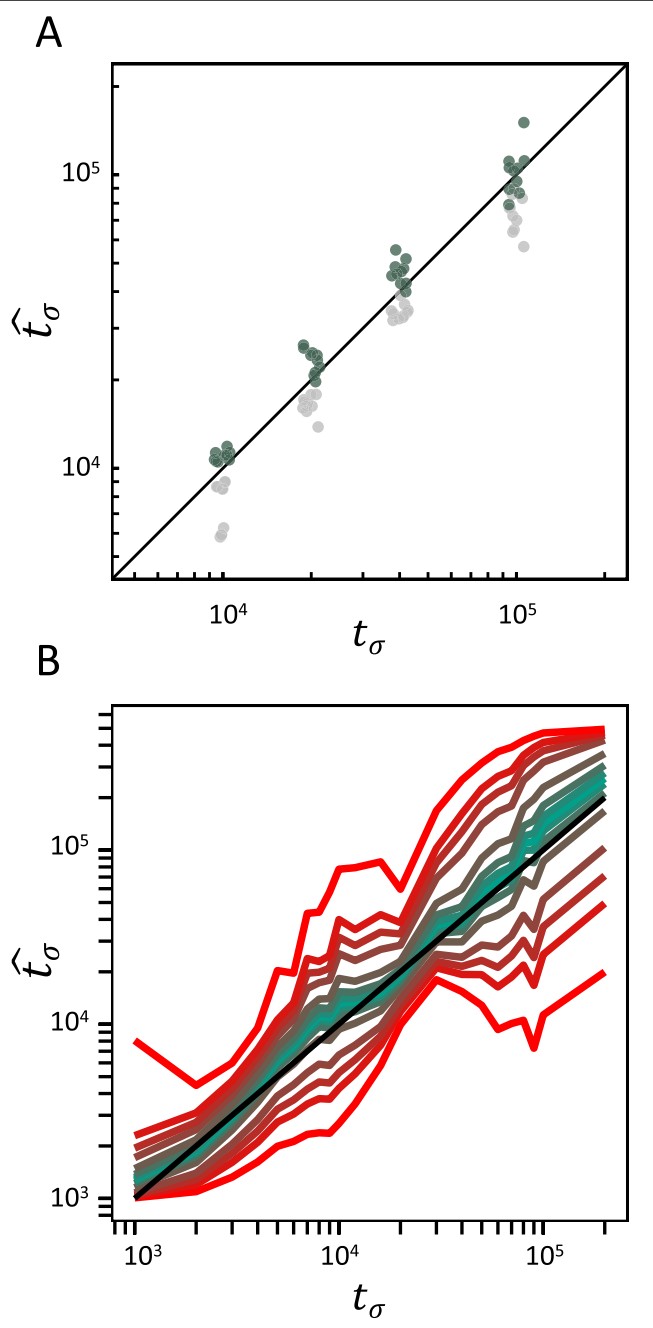

**Figure 4.** Accuracy of *teSMC* and *tsABC* in the presence of background selection (BGS). Inference of times of transition from outcrossing (σ=0.1) to predominant selfing (σ=0.99) using (**A**) *teSMC* and (**B**) *tsABC*. Simulations were done under constant population size and negative selection acting on exonic sequences. The spatial distribution of exonic sequences was fixed and taken from the annotation of *A. thaliana*. Negative selection was modeled using the distribution of fitness effects from **Hämälä and Tiffin, 2020**. (**A**) Comparison between simulated values of $t_\sigma$ and estimates obtained with *teSMC* using the one-transition mode. Estimations were conducted with and without masking exonic sequences subject to negative selection. The inference process was repeated 10 times for each experimental condition, employing independently simulated data sets. (**B**) Same analyses as in panel A but conducted with *tsABC*. Except for selection, simulations were done as in **Figure 3H**. Colored lines represent the average quantiles for 100 posterior distributions obtained with *tsABC*.

The online version of this article includes the following figure supplement(s) for figure 4:

**Figure supplement 1.** Accuracy of *tsABC* in the presence of background selection (BGS).

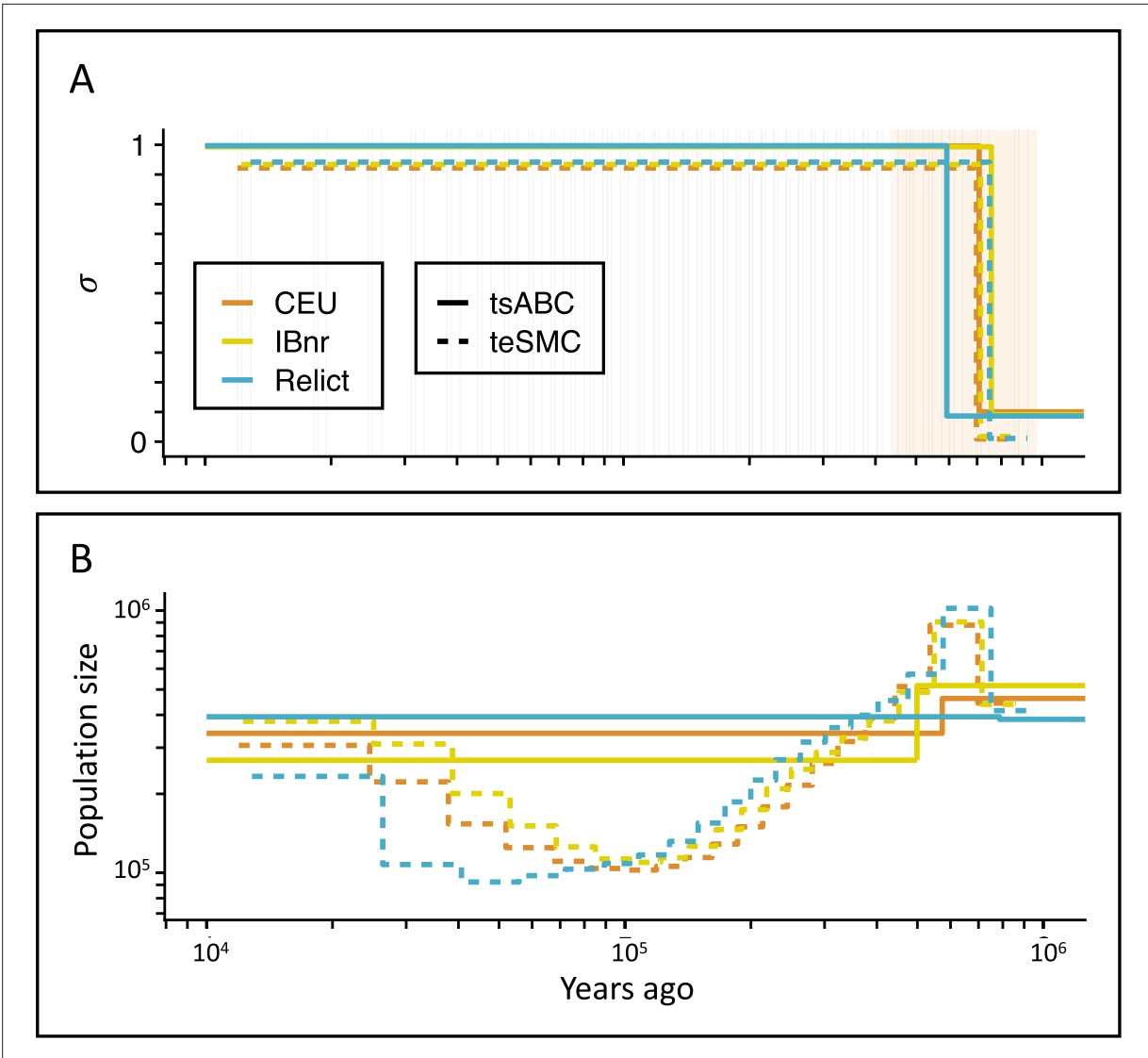

**Figure 5.** Inference of the time of transition from outcrossing to selfing in *A. thaliana*. (**A**) Inferred transitions from outcrossing to selfing for three independent genetic clusters of *A. thaliana* from the 1001 genomes project (CEU, IBnr, Relict) using *tsABC* and *teSMC*. The 95% CI (CEU) of the posterior distribution of $t_\sigma$ (*tsABC*) is indicated in light-orange. (**B**) Co-estimated population sizes over time with a single population change (*tsABC*) or piecewise constant (*teSMC*). 20 samples (CEU, IBnr for *teSMC*), 17 samples (Relicts), 99 samples (CEU, *tsABC*), or 66 samples (IBnr, *tsABC*) were used.

The online version of this article includes the following figure supplement(s) for figure 5:

**Figure supplement 1.** Genomic regions of *A. thaliana* genome (TAIR10) used for the *teSMC* and *tsABC* analyses.

**Figure supplement 2.** Inference of the time of transition from outcrossing to selfing in *A. thaliana* using two different sets of genomic regions.

sizes inferred by *teSMC* are comparable to the ones obtained by ***Durvasula et al., 2017*** and ***Fulgione and Hancock, 2018*** between present and 400,000 generations in the past; pending that their *msmc2* results are rescaled by 1/(1+*F*), which is approximately two in this case. For times older than 400,000, around the age of the transition, ***Durvasula et al., 2017*** reports smaller effective sizes than *teSMC*.

**Table 1.** Estimated times of transitions from predominant outcrossing to predominant selfing in *A. thaliana*.

Estimations were conducted for three different ancestry groups: central Europe (CEU), Iberian non-relicts (IBnr), and Relicts using both *teSMC* and *tsABC*. The 95% CI of all jointly inferred parameters are provided in *Supplementary file 1*, Table S6. For both methods, polymorphism data was measured on five genomic regions of 1 Mb, located on the five chromosomes of *A. thaliana*. For *teSMC* exonic regions were excluded from the analysis. Only accessions with cluster membership >95% were included.

| Method | Population | Sample size | Mode | 95% credibility interval | |
|--------|-----------|-------------|------|--------------------------|---|
| *teSMC* | CEU | 20 | 697,490 | *NA* | *NA* |
| *teSMC* | IBnr | 20 | 713,421 | *NA* | *NA* |
| *teSMC* | Relicts | 17 | 749,668 | *NA* | *NA* |
| *tsABC* | CEU | 99 | 707,995 | 443,486 | 973,841 |
| *tsABC* | IBnr | 66 | 756,976 | 397,049 | 988,708 |
| *tsABC* | Relicts | 17 | 592,321 | 386,406 | 934,499 |

## Discussion

While the biological importance of shifts in mating systems or reproductive modes is long recognized as a key evolutionary and ecological process, no method based on genome-wide polymorphism data was available to estimate the age of a change in reproductive mode, while accounting for demographic history. In this study we show that by accounting for both the frequency distribution of SNPs and the distribution of historical recombination events, it is possible to recover shifts in reproductive modes based on a small number of fully sequenced genomes. The key idea underlying our approach is to leverage the properties of genomic segments delimited by recombination events ($T_{MRCA}$-segments). Our simulations show that the molecular signature of a transition to selfing is twofold. The first effect is a change in the relation between the age and the length of genomic segments that occurs at the time of the transition (*Figure 1*, *Figure 1—figure supplements 1–3*), and the second is a characteristic spatial distribution of the segments along the chromosome, where segments older than the transition tend to occur as clusters (*Figure 1C–E*). Several SMC-based methods already take advantage of both the distribution of age and lengths of $T_{MRCA}$-segments to estimate past demographic history (*Li and Durbin, 2011*; *V Barroso et al., 2019*; *Steinrücken et al., 2019*), spatial structure (*Wang et al., 2020*), heterogeneous recombination rates along the genome (*V Barroso et al., 2019*), or life-history traits such as dormancy (*Sellinger et al., 2020*). SMC-based as well as other inference methods (*Kerdoncuff et al., 2020*) rely on estimating changes in the ratio of population recombination rate by population mutation rate along the genome (the ratio $\rho/\theta$). Yet, none of them attempted to leverage the information on the distribution of $T_{MRCA}$-segments to also estimate temporal changes of the ratio $\rho/\theta$, which in our case amounts to estimate variation of the selfing rate. Interestingly, as most inference methods are developed with a main application to human data in mind, the estimation of a changing $\rho/\theta$ ratio has not yet been a relevant and pressing question. *Deng et al., 2021*, described how temporal recombination rates could be estimated from reconstructed ARGs, but only evaluated the performance of this approach on simulations with a constant population size and rate.

In this study we develop two approaches to estimate change in selfing in time: one building upon the Markovian assumption of coalescence events along the genome (*teSMC*) and another one by considering the dependencies between the genetic diversity observed in successive genomic windows of fixed size (*tsABC*). The advantage of the *teSMC* approach is that it is most effective in identifying the boundaries and the ages of $T_{MRCA}$-segments and therefore allows capturing the characteristic effect of a shift to selfing (as shown in *Figure 1A and B*). *teSMC* is also efficient in the way it optimizes the likelihood such that the method can be executed on a single desktop computer when applied to empirical data. The drawback of the *teSMC* method is that extending it to more complicated

demographic scenario is mathematically and computationally difficult (e.g. to selection or admixture). Conversely, the *tsABC* approach can easily be extended to more complicated demographic and non-neutral scenarios but suffers from a lower statistical performance caused by the approximation made in the summarization of the data; and by the requirement to conduct many simulations that can only be realistically obtained on a high-performance computing cluster. Our results suggest our approaches to be robust to the presence of BGS, and to a lesser extent to variation of recombination and mutation rate along the genome. We nevertheless show that a spurious change in selfing rate is not generated by variation in recombination rate along the genome (*Figure 2—figure supplement 6*). We finally highlight the novelty in the design of the *tsABC* approach, which introduces a new summary statistic, the transition matrix for heterozygosity levels, which contains similar information as in the transition matrix computed by the *teSMC* method and which could be incorporated as a useful summary statistics in other ABC methods. Both methods belong thus to the same conceptual framework and are complementary.

Our results indicate that genomic distributions of $T_{MRCA}$ estimates are more informative about the age of transitions to self-fertilization than approaches relying on the rate of pseudogenization of SI loci (*Bechsgaard et al., 2006*). Furthermore, *S-locus*-based inferences of shifts to selfing are limited to species for which such transitions have been caused by a loss-of-function mutation in the *S-locus* and for which the *S-locus* has been identified and properly assembled. This information is only partially available for other plant species, and the genetic determinism of selfing also varies between genera (*Franklin-Tong, 2008*). Thus, our inference methods offer the opportunity to test for the existence and the timing of changes in mode of reproduction in potentially any species providing full-genome polymorphism data are available. They also pave the way for addressing long-standing questions on the evolution of reproductive systems which cannot be directly tested from field experimental or even phylogenetic approaches. How frequent and how recent are transitions from outcrossing to selfing is the main question, and the one that motivated our work. Phylogenetic methods can at best infer that a transition has occurred at some time on a given branch of a phylogeny. For example, considering the case of *A. thaliana*, knowing that the two sister species, *A. lyrata* and *Arabidopsis halleri*, are self-incompatible indicates that the shift to selfing occurred after the divergence between the two lineages, that is between present time and 13 millions years (*Beilstein et al., 2010*), which is poorly informative. In addition, from phylogenetic character mapping, when two or more sister species share the same character state, most of the time the shift to this state is inferred before the divergence of the species. However, it may not be the case for very labile traits with high extinction rate as self-fertilization. In the fungus genus *Neurospora*, although a clade of species shares a homothallic mating system (equivalent to selfing), a detailed molecular analysis of the *mat* locus that controls mating system revealed that the breakdown of this locus occurred several times independently (*Gioti et al., 2012*). Reversion from selfing to outcrossing is supposed to be very rare but this question is still debated (*Barrett, 2013*). Our methods provide the adequate tool to tackle this question and their systematic application to various species may help discovering such possible reversion or even more complex scenarios. As an example, we apply *teSMC* to estimate scenarios of transition from outcrossing to selfing followed by reversion, or the reverse, as well as more gradual (stepwise) transitions (*Figure 2—figure supplements 7–8*). The method performed well and is thus promising for detecting more complex histories.

Another useful application of the methods is for demographic inferences when shifts in mating systems are suspected. As they alter the distribution of age and lengths of $T_{MRCA}$-segments, they can lead to spurious shifts in population size if not taken into account (*Figure 2—figure supplement 3*). For example, using an SMC approach, ancestral drops in population sizes have been inferred in the selfers *Capsella orientalis* and *C. bursa-pastoris* but not in the outcrosser *Coreopsis grandiflora* (*Kryvokhyzha et al., 2019*). This could correspond to real changes in population size but also to transition toward selfing. This issue is especially worth being considered in cultivated species. The demographic history associated with domestication is a central question for the study of crop species and demographic scenarios have been inferred in many species. However, shifts, or at least variations, in mating system occurred quite frequently during plant domestication (e.g. African rice, tomato, grapevine, melon) (*Glémin and Bataillon, 2009*; *Meyer et al., 2012*) and taking such variations into account would help refining demographic scenarios.

The methods we developed focus on outcrossing/selfing transitions. However, they could be extended to other reproductive modes such as sex/asex transitions by adapting the relevant population parameters using existing works on coalescence with facultative sexual reproduction (*Hartfield et al., 2018*). Overall, our methods may open new ways to answer the old riddle of why so many species do reproduce sexually (*Barton and Charlesworth, 1998*).

## Materials and methods
### Modeling of a transition to selfing using forward and coalescent simulations

To model a transition from outcrossing to predominant selfing, we considered a single population composed of $N$ diploid individuals. At each generation, each offspring is generated by self-fertilization of a single individual or by outcrossing with probabilities $\sigma$ and $1-\sigma$, respectively, where $\sigma$ is the selfing rate. Unless stated otherwise, transitions to predominant selfing were modeled by allowing the selfing rate to change instantaneously from $\sigma_{ANC}$ to $\sigma_{PRES}$ at time $t_\sigma$. The mutation and recombination were set to $1 \times 10^{-8}$ events per generation per nucleotide. When needed the population size was allowed to change instantaneously from $N_{ANC}$ to $N_{PRES}$ at $t_N$. This model was implemented using the WF simulation mode in *slim3* (*Haller and Messer, 2019*); scripts to simulate genetic data using this model are available on our git repository (https://github.com/laurentlab-mpipz/struett_and_sellinger_et_al.git). We used this model to generate genetic variation for a sample of $n$=20 haploid genomes, sampled from 20 different individuals, and composed of five DNA sequences of 1 Mb each.

The same model was implemented in a coalescent framework using *msprime* (*Kelleher et al., 2016*). Following *Nelson et al., 2020*, who showed that continuous-time coalescent simulations of large sequences cause biases in patterns of identity-by-descent and LD, we implemented a hybrid model in which the first 1000 generations were simulated using a discrete-time coalescent process and the following generations were modeled using the SMC' algorithm (*Marjoram and Wall, 2006*). The coalescent implementation, which runs significantly faster than the forward WF implementation, was used for the ABC and performance analyses (see below), while the WF forward implementation was used to assess the quality of the coalescent-with-selfing approximation proposed by *Nordborg and Donnelly, 1997*; *Nordborg, 2000*; *Figure 1—figure supplement 1*, and was extended to study the consequences of negative selection on inference with the *tsABC* and *teSMC* methods (see below).

### Analysis of $T_{MRCA}$-segments in simulated data

For the forward and coalescent simulations, we wrote functions to obtain the lengths and the time to the most recent ancestor ($T_{MRCA}$) of $T_{MRCA}$-segments. $T_{MRCA}$-segments are sets of contiguous nucleotides in a sample of size two that share the same MRCA. The joint distribution of $T_{MRCA}$ and lengths of those segments (TL-distributions) was used to describe the consequences of a transition to selfing at the genomic level and how it differs from a change in population size (*Figure 1*, *Figure 1—figure supplement 1*, *Figure 1—figure supplement 3*). $T_{MRCA}$-segments were analyzed by identifying consequential recombination events in the history of the sample (i.e. events that lead to the inclusion of a new MRCA) and the corresponding breakpoints represented the boundaries of the successive segments. Then, the $T_{MRCA}$ of each segment was obtained by identifying the MRCA of each segment.

We also calculated the transition matrix of $T_{MRCA}$ of successive $T_{MRCA}$-segments along the genome (TM$_{true}$, *Figure 1D and E*). For this, we discretized $T_{MRCA}$ values and counted the frequencies of segment transitions along simulated sequences for each combination of discrete $T_{MRCA}$ values. $T_{MRCA}$ were discretized using a similar approach as in MSMC (*Schiffels and Durbin, 2014*) with the lower boundary of bin $i$ given by, $-8N \times \log(1-i/m)$, where $N$ is the population size and $m$ the total number of bins. The relevant code can be found at https://github.com/laurentlab-mpipz/struett_and_sellinger_et_al.git, (*Sellinger, 2023*; *Strütt, 2023*).

## Calculation of summary statistics of polymorphism data

While TL-distributions carry the characteristic signature of shifts to selfing, they are also challenging to infer from empirical genetic data. Therefore, we used three summarization approaches to capture this signal using polymorphism data: (1) The unfolded SFS, which is the distribution of absolute derived allele frequencies in the sample and is known to carry information about past population-size changes. (2) A discretized distribution of LD decay inspired from the approach taken by *Boitard et al., 2016*, who used it jointly with the SFS to estimate past changes in population sizes. Unlike the SFS, which only carries information about $N$ but not the recombination rate ($r$), LD-decay depends on the product of $N$ and $r$. Combining both distributions therefore allows to capture the signature of changes in $N$ and $r$. LD was calculated as $r^2$ from a subset of 10,000 randomly chosen SNPs and discretized into discrete physical distances with following breakpoints: 6105; 11,379; 21,209; 39,531; 73,680; 137,328; 255,958; 477,066; 889,175 bp. (3) Window-based transition matrix ($TM_{win}$): While $TM_{true}$ carries a characteristic signal to estimate shifts to selfing, it is not straightforward to calculate it using polymorphism data. This is because the boundaries of $T_{MRCA}$-segments are not directly observable and need to be inferred themselves. $TM_{win}$ captures some of the information in $TM_{true}$ by computing the pairwise diversity in non-overlapping successive windows of 10 kb for a sample of size two.

## Simulations with BGS

Simulations with BGS were conducted with *slim3*. We used the DFE estimated by *DFEalpha* for *A. thaliana* published by *Hämälä and Tiffin, 2020*. The DFE was used to assign negative selection coefficients to simulated coding non-synonymous genetic variants only (i.e. we did not simulate negative selection on functional non-coding regions). We took care of simulating realistic proportions and spatial distributions of coding sequences by using the positional information of CDS from the annotation of the reference genome of *A. thaliana* (*Arabidopsis Genome Initiative, 2000*). Except for the DFEs and genetic structure, all other parameters and dataset dimensions were identical to the simulations without negative selection.

## Application to *A. thaliana*

To calculate summary statistics on the *A. thaliana* dataset, we used the imputed genotype matrix provided on the 1001 genomes website (https://1001genomes.org/data/GMI-MPI/releases/v3.1/SNP_matrix_imputed_hdf5/). We used samples from three separate genetic clusters: central European (CEU), Iberian non-relicts (IBnr), and Relicts. Only accessions with cluster membership >95% were included; because our model does not account for gene flow between ancestry groups (*Supplementary file 1*, Table S4). The final sample sizes are provided in *Table 1*. We conducted a genome-wide sliding-window analysis of nucleotide diversity with *vcftools* (*Danecek et al., 2011*) and identified large genomic regions with elevated nucleotide diversity, centered around pericentromeric regions (*Underwood et al., 2018*). Estimations were conducted using five 1-Mb-long genomic regions (*Supplementary file 1*, Table S5) that did not overlap with these elevated diversity regions (*Figure 5—figure supplement 1*). We resampled 12 haplotypes multiple times and calculated the combined summary statistics, SFS, LD, and $TM_{win}$. We centralized and normalized the statistics and calculated the first 20-PLS (see Appendix 2). *tsABC* was conducted on the *A. thaliana* data, by using 12 samples for 5 independent loci of 1 Mb length. The mutation and recombination rates were set to 6.95e-9 (*Ossowski et al., 2010*) and 3.6e-9. The recombination rate is the genome-wide average provided by *Salomé et al., 2012*. We simulated a total set of 130,000 vectors of summary statistics. Parameter estimates were conducted as described for the ABC performance analysis. We provided the mode of the average posterior distributions as a final result. In addition, we used a subsample of 20 *A. thaliana* sequences per genetic cluster to estimate demography and the transition to selfing using *teSMC* in the one-transition mode. Data and scripts can be found at https://github.com/laurentlab-mpipz/struett_and_sellinger_et_al.git and https://github.com/TPPSellinger/eSMC2.

## Data availability

A complete detailed description of the *teSMC* and *tsABC* methods is in the SI text Appendices 1 and 2. The *tsABC* method can be found on GitHub (https://github.com/sstruett/tsABC). Scripts for all figures, simulations, and the specific *tsABC* workflow used in this study can be found on a separate repository (https://github.com/laurentlab-mpipz/struett_and_sellinger_et_al.git). The *teSMC* method

and a tutorial to simulate and analyze data are available on GitHub (https://github.com/TPPSellinger/eSMC2).

## Acknowledgements

This work was funded by the Deutsche Forschungsgemeinschaft (DFG, German Research Foundation), project 462181533 to SL and projects 317616126, 254587930, and 447121532 to AT.

## Additional information

### Funding

| Funder | Grant reference number | Author |
|---|---|---|
| Max Planck Institute for Plant Breeding Research | open access funding | Stefan Strütt Stefan Laurent |

### Author contributions

Stefan Strütt, Thibaut Sellinger, Conceptualization, Software, Formal analysis, Validation, Investigation, Visualization, Methodology, Writing - original draft; Sylvain Glémin, Supervision, Methodology, Writing - original draft, Writing - review and editing; Aurélien Tellier, Conceptualization, Supervision, Funding acquisition, Investigation, Methodology, Writing - original draft, Writing - review and editing; Stefan Laurent, Conceptualization, Supervision, Funding acquisition, Investigation, Methodology, Writing - original draft, Project administration, Writing - review and editing

### Author ORCIDs
Stefan Strütt ⓘ http://orcid.org/0000-0002-2785-2815
Sylvain Glémin ⓘ http://orcid.org/0000-0001-7260-4573
Stefan Laurent ⓘ http://orcid.org/0000-0003-4016-5427

### Decision letter and Author response
Decision letter https://doi.org/10.7554/eLife.82384.sa1
Author response https://doi.org/10.7554/eLife.82384.sa2

## Additional files

### Supplementary files
• Supplementary file 1. Supplementary tables. Parameters for simulated datasets to investigate the performance of *tsABC*. Table S2: Parameter priors used for the performance analysis of *tsABC*. Table S3: Parameter priors used to estimate the transition from outcrossing to selfing in *A. thaliana*. Table S4: *A. thaliana* samples with >95% cluster membership (https://1001genomes.github.io/admixture-map/) used to estimate the transition from outcrossing to selfing obtained from the 1001 genomes website (https://1001genomes.org/data/GMI-MPI/releases/v3.1/SNP_matrix_imputed_hdf5/1001_SNP_MATRIX.tar.gz). Table S5: Genomic regions of *A. thaliana* in TAIR10 used for the *tsABC* and *teSMC* analyses. Table S6: Jointly estimated parameters in the context of transitions from predominant outcrossing to predominant selfing in *A. thaliana*.

• MDAR checklist

### Data availability
A complete detailed description of the teSMC and tsABC methods is in the SI text Appendix 1 and 2. The tsABC method can be found on GitHub (https://github.com/sstruett/tsABC, copy archived at

swh:1:rev:370866ae9f0de08582c86f34edbdf3193ca7bb2c). Scripts for all figures, simulations, and the specific tsABC workflow used in this study can be found on a separate repository (https://github.com/laurentlab-mpipz/struett_and_sellinger_et_al.git). copy archived at The teSMC method and a tutorial to simulate and analyze data are available on GitHub (https://github.com/TPPSellinger/eSMC2, copy archived at swh:1:rev:3cb705d7fc9f2e0918bb488eec1da30d4e08a2ec).

The following previously published dataset was used:

| Author(s) | Year | Dataset title | Dataset URL | Database and Identifier |
|---|---|---|---|---|
| The 1001 Genomes Consortium | 2016 | 1,135 Genomes Reveal the Global Pattern of Polymorphism in *Arabidopsis thaliana* | http://1001genomes.org/data/GMI-MPI/releases/v3.1/SNP_matrix_imputed_hdf5/1001_SNP_MATRIX.tar.gz | 1001genomes, 10.1016/j.cell.2016.05.063 |

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

# Appendix 1

## Description of teSMC
### teSMC

Note that teSMC also includes the effect of dormancy (or seed banks) although not discussed in the manuscript. To define our HMM we need to define:

> The hidden states,
> The signal (observed data) to be used as input,
> A transition matrix (probability of switching from one state to another),
> An emission matrix (probability of observing the data given the current hidden state),
> An initial probability (probability of hidden states at the first position of the sequence).

### Notations and assumptions

We here define the different notations used and their meaning:

> $\sigma_t$: self fertilization rate (between 0 and 1) at time $t$,
> $\beta_t$: dormancy rate (between 0 and 1) at time $t$,
> $r_t$: recombination rate per nucleotide (at time $t$) per units of coalescence time,
> $\mu$: mutation rate per nucleotide per units of coalescence time,
> $\mu_b$: ratio of mutation rate during the dormant stage over the mutation rate during the active stage,
> $u$: time at which the recombination occurs (follows a piecewise uniform distribution),
> $L$: sequence length in bp,
> $N_t$: population size at time $t$,
> $\chi_t$: scaling factor for the population size at time $t$ ($N_t = \chi_t N_e$).

The assumptions of the model are:

> Piecewise constant population size,
> Piecewise constant selfing, dormancy, and recombination rate in time,
> Constant mutation rate in time,
> Constant mutation and recombination rate along the sequence.

### Observations

Our observations, also called signal, is a sequence of 1 and 0. This sequence is build from phasing the DNA sequences of two individuals. When going along the sequence, at a given position, if both nucleotides are similar then the signal is 0 (no mutation occurred). If both nucleotides are different, then a mutation occurred, and the signal is 1.

### Hidden states

We define our hidden states at one position on the genome as the coalescent time between the two individuals. This is denoted as the coalescent time $t$ ($t>0$). More precisely, because the state space must be finite, we discretize time in several ($n$) intervals. At one interval, the hidden state takes a value $\alpha$ if $t \in [T_\alpha, T_{\alpha+1}]$, where $\alpha \in [0, (n-1)]$. We define $T_\alpha$ as:

$$T_\alpha = -\frac{(2 - \sigma_0)}{2\beta_0^2} \ln(1 - \frac{\alpha}{n}). \tag{1}$$

In addition, a transition from a coalescent time $s$ to time $t$ ($t \neq s$) at the next sequence position can only occur if a recombination has happened in-between the two positions. We describe this event below.

## Initial probability

We use the equilibrium probability as initial probability. The equilibrium probability is defined as the probability that the first coalescent happens at each time interval and is thus given by:

$$
\begin{aligned}
q_o(\alpha) &= \int_{T_\alpha}^{T_{\alpha+1}} \frac{2\beta_\alpha^2}{(2 - \sigma_\alpha)\chi_\alpha} e^{\int_0^t \frac{-2\beta_v^2}{(2 - \sigma_v)\chi_v} dv} \, dt \\
q_o(\alpha) &= \int_{T_\alpha}^{T_{\alpha+1}} \frac{2\beta_\alpha^2}{(2 - \sigma_\alpha)\chi_\alpha} e^{\int_0^{T_\alpha} \frac{-2\beta_v^2}{(2 - \sigma_v)\chi_v} dv} e^{\int_{T_\alpha}^{t} \frac{-2\beta_v^2}{(2 - \sigma_v)\chi_v} dv} \, dt \\
q_o(\alpha) &= e^{\int_0^{T_\alpha} \frac{-2\beta_v^2}{(2 - \sigma_v)\chi_v} dv} \int_{T_\alpha}^{T_{\alpha+1}} \frac{2\beta_\alpha^2}{(2 - \sigma_\alpha)\chi_\alpha} e^{\frac{-2(t - T_\alpha)\beta_\alpha^2}{(2 - \sigma_\alpha)\chi_\alpha}} \, dt \\
q_o(\alpha) &= e^{\sum_{\eta=0}^{\alpha-1} \frac{-2\beta_\eta^2}{(2 - \sigma_\eta)\chi_\eta} \Delta_\eta} (1 - e^{\frac{-2\Delta_\alpha \beta_\alpha^2}{(2 - \sigma_\alpha)\chi_\alpha}})
\end{aligned}
\tag{2}
$$

## Transition matrix

A transition to state $t$ from state $s$ ($t \neq s$) only occurs if there is a recombination event. Because recombination, dormancy, and selfing are assumed non-constant through time, recombination events along the coalescence tree follow an inhomogeneous Poisson process. Hence, the probability for a recombination to occur conditioned to the current coalescence time $s$ is:

$$P(rec|s) = \left( 1 - e^{-\int_0^s \frac{2(1 - \sigma_k)\beta_k}{2 - \sigma_k} 2r_k dk} \right) \tag{3}$$

We assume that a recombination event occurred at time $u$ ($u < s$), where $u$ follows a piecewise uniform distribution between 0 and $s$, that is uniform in each hidden state but the density between hidden state can change. Then, we can define three possible scenarios, namely the new coalescent time of the coalescent tree can be smaller ($t < s$), bigger ($t > s$), or unchanged ($t = s$) compared to the initial tree. These computations are also found in previous PSMC approaches (**Li and Durbin, 2011**) and in **Sellinger et al., 2020**.

### t<s

The resulting floating branch of the recombination event coalesces at time $t < s$. This means that the coalescence did not occur before time $t$ (including itself). In addition we have $u < t$. The transition probability is therefore obtained as:

$$P(t|s, u) = \frac{2\beta_t^2}{(2 - \sigma_t)\chi_t} (e^{\int_u^t -\frac{4\beta_v^2}{(2 - \sigma_v)\chi_v} dv}) \tag{4}$$

### t=s

The resulting floating branch of the recombination event (self) coalesces before time $t$. We therefore obtain the transition probability:

$$P(s|s, u) = \int_u^s \frac{2\beta_k^2}{(2 - \sigma_k)\chi_k} e^{\int_u^k -\frac{4\beta_v^2}{(2 - \sigma_v)\chi_v} dv} dk \tag{5}$$

### t>s

The resulting floating branch of the recombination event does not coalesce before time $s$, and no coalescent event happens before time $t$. We therefore obtain the transition probability:

$$P(t|s, u) = \frac{2\beta_t^2}{(2 - \sigma_t)\chi_t} e^{\int_u^s -\frac{4\beta_v^2}{(2 - \sigma_v)\chi_v} dv} e^{\int_s^t -\frac{2\beta_v^2}{(2 - \sigma_v)\chi_v} dv} \tag{6}$$

## Transition probability in continuous time

The transition probability conditioned to the occurrence of a recombination event occurring at time can be summarized as:

$$p(t|s, u) = \begin{cases} \frac{2\beta_t^2}{(2 - \sigma_t)\chi_t}(e^{\int_u^t -\frac{4\beta_v^2}{(2 - \sigma_v)\chi_v} dv}) & if \quad u < t < s \\ \int_u^s \frac{2\beta_k^2}{(2 - \sigma_k)\chi_k} e^{\int_u^k -\frac{4\beta_v^2}{(2 - \sigma_v)\chi_v} dv} dk & if \quad t = s \\ \frac{2\beta_t^2}{(2 - \sigma_t)\chi_t} e^{\int_u^s -\frac{4\beta_v^2}{(2 - \sigma_v)\chi_v} dv} e^{\int_s^t -\frac{2\beta_v^2}{(2 - \sigma_v)\chi_v} dv} & if \quad t > s \\ 0 & if \quad otherwise \end{cases} \tag{7}$$

If all $\sigma_k = 0$, $\beta_k = 1$, we fall back on the probability from PSMC'. To find $p(t|s)$ we use the total probability formula which is given as:

$$p(t|s) = \int_0^s p(u)p(t|s, u)du \tag{8}$$

As explained above, time is discretized in $n$ small intervals. The hidden state is $\alpha$ if $t \in [T_\alpha, T_{\alpha+1}]$, where $\alpha \in [0, (n - 1)]$ with $T_\alpha$ defined as:

$$T_\alpha = -\frac{(2 - \sigma_0)}{2\beta_0^2} \ln(1 - \frac{\alpha}{n}) \tag{9}$$

The transition matrix summarizes the probability of transition from one hidden state to another. Therefore, we need to compute the probability of the coalescent time at the previous sequence position (which is here $s$) to belong to the state $\gamma$. To do this we replace $s$ by the expected coalescent time knowing the coalescence events to be occurring in the hidden state $\gamma$. We write this expected value as $t_\gamma$ which is defined as:

$$\begin{aligned} t_\gamma = E[\text{Coalescent time}|\gamma] &= \frac{E[\text{Coalescent time} \cap \gamma]}{P(\gamma)} = \frac{\int_{T_\gamma}^{T_{\gamma+1}} t\Lambda_\gamma e^{-\int_0^t \Lambda_v dv} dt}{q_0(\gamma)} \\ &= \frac{\Lambda_\gamma \int_{T_\gamma}^{T_{\gamma+1}} te^{-\int_0^{T_\gamma} \Lambda_v dv} e^{-\int_{T_\gamma}^t \Lambda_v dv} dt}{q_0(\gamma)} = \frac{\Lambda_\gamma e^{-\int_0^{T_\gamma} \Lambda_v dv} \int_{T_\gamma}^{T_{\gamma+1}} te^{-\int_{T_\gamma}^t \Lambda_v dv} dt}{q_0(\gamma)} \\ &= \frac{\Lambda_\gamma \int_{T_\gamma}^{T_{\gamma+1}} te^{(T_\gamma - t)\Lambda_\gamma} dt}{(1 - e^{-\Delta_\gamma \Lambda_\gamma})} = \frac{T_\gamma - T_{\gamma+1} e^{-\Delta_\gamma \Lambda_\gamma}}{(1 - e^{-\Delta_\gamma \Lambda_\gamma})} + \frac{\int_{T_\gamma}^{T_{\gamma+1}} e^{(T_\gamma - t)\Lambda_\gamma} dt}{(1 - e^{-\Delta_\gamma \Lambda_\gamma})} \\ &= \frac{T_\gamma - T_{\gamma+1} e^{-\Delta_\gamma \Lambda_\gamma}}{(1 - e^{-\Delta_\gamma \Lambda_\gamma})} + \frac{(1 - e^{-\Delta_\gamma \Lambda_\gamma})}{\Lambda_\gamma(1 - e^{-\Delta_\gamma \Lambda_\gamma})} = \frac{T_\gamma - T_{\gamma+1} e^{-\Delta_\gamma \Lambda_\gamma}}{(1 - e^{-\Delta_\gamma \Lambda_\gamma})} + \frac{1}{\Lambda_\gamma} \end{aligned} \tag{10}$$

With the following definitions:

$$\begin{aligned} \Delta_\gamma &= T_{\gamma+1} - T_\gamma \\ \Lambda_\gamma &= \frac{2\beta_\gamma^2}{(2 - \sigma_\gamma)\chi_\gamma} \end{aligned} \tag{11}$$

From the previous equation, we can define our transition probability:

$$p(\alpha|\gamma) = \int_{T_\alpha}^{T_{\alpha+1}} p(t|t_\gamma)dt = \int_{T_\alpha}^{T_{\alpha+1}} \int_0^{t_\gamma} p(u)p(t|t_\gamma, u)dudt \tag{12}$$

## Calculation of $p(\alpha|\gamma)$

To compute $p(\alpha|\gamma)$, we first need $p(t|t_\gamma)$, which is obtained under three possible cases.

Case 1: $\alpha < \gamma$

$$
\begin{aligned}
p(t|t_\gamma) &= P_\gamma \int_0^t \frac{\pi_u \frac{2\beta_t^2}{(2-\sigma_t)\chi_t}(e^{\int_u^t -\frac{4\beta_v^2}{(2-\sigma_v)\chi_v}dv})}{\Pi_\gamma} du \\
&= P_\gamma \Big( \sum_{\eta=1}^{\alpha-1} \int_{T_\eta}^{T_{\eta+1}} \frac{\pi_u \frac{2\beta_t^2}{(2-\sigma_t)\chi_t}(e^{\int_u^t -\frac{4\beta_v^2}{(2-\sigma_v)\chi_v}dv})}{\Pi_\gamma} du \\
&\quad + \int_{T_\alpha}^t \frac{\pi_u \frac{2\beta_t^2}{(2-\sigma_t)\chi_t}(e^{\int_u^t -\frac{4\beta_v^2}{(2-\sigma_v)\chi_v}dv})}{\Pi_\gamma} du \Big) \\
&= P_\gamma \Big( \sum_{\eta=1}^{\alpha-1} \int_{T_\eta}^{T_{\eta+1}} \frac{\pi_\eta \frac{2\beta_t^2}{(2-\sigma_t)\chi_t}(e^{\int_u^t -\frac{4\beta_v^2}{(2-\sigma_v)\chi_v}dv})}{\Pi_\gamma} du \\
&\quad + \int_{T_\alpha}^t \frac{\pi_\alpha \frac{2\beta_t^2}{(2-\sigma_t)\chi_t}(e^{\int_u^t -\frac{4\beta_v^2}{(2-\sigma_v)\chi_v}dv})}{\Pi_\gamma} du \Big) \\
&= P_\gamma \Big( \sum_{\eta=1}^{\alpha-1} \int_{T_\eta}^{T_{\eta+1}} \frac{\pi_\eta \frac{2\beta_\alpha^2}{(2-\sigma_\alpha)\chi_\alpha}(e^{\int_u^{T_{\eta+1}} -\frac{4\beta_v^2}{(2-\sigma_v)\chi_v}dv})(e^{\int_{T_{\eta+1}}^t -\frac{4\beta_v^2}{(2-\sigma_v)\chi_v}dv})}{\Pi_\gamma} du \\
&\quad + \int_{T_\alpha}^t \frac{\pi_\alpha \frac{2\beta_\alpha^2}{(2-\sigma_\alpha)\chi_\alpha}(e^{\int_u^t -\frac{4\beta_v^2}{(2-\sigma_v)\chi_v}dv})}{\Pi_\gamma} du \Big) \\
&= P_\gamma \Big( \sum_{\eta=1}^{\alpha-1} \int_{T_\eta}^{T_{\eta+1}} \frac{\pi_\eta \frac{2\beta_\alpha^2}{(2-\sigma_\alpha)\chi_\alpha}(e^{-(T_{\eta+1}-u)\frac{4\beta_\eta^2}{(2-\sigma_\eta)\chi_\eta}})(e^{\int_{T_{\eta+1}}^t -\frac{4\beta_v^2}{(2-\sigma_v)\chi_v}dv})}{\Pi_\gamma} du \\
&\quad + \int_{T_\alpha}^t \frac{\pi_\alpha \frac{2\beta_\alpha^2}{(2-\sigma_\alpha)\chi_\alpha}(e^{-(t-u)\frac{4\beta_\alpha^2}{(2-\sigma_\alpha)\chi_\alpha}})}{\Pi_\gamma} du \Big) \\
&= P_\gamma \Big( \sum_{\eta=1}^{\alpha-1} \frac{\pi_\eta \frac{2\beta_\alpha^2}{(2-\sigma_\alpha)\chi_\alpha}(1 - e^{-\Delta_\eta \frac{4\beta_\eta^2}{(2-\sigma_\eta)\chi_\eta}})(e^{\int_{T_{\eta+1}}^t -\frac{4\beta_v^2}{(2-\sigma_v)\chi_v}dv})}{\frac{4\beta_\eta^2}{(2-\sigma_\eta)\chi_\eta}\Pi_\gamma} \\
&\quad + \frac{\pi_\alpha \frac{2\beta_\alpha^2}{(2-\sigma_\alpha)\chi_\alpha}(1 - e^{-(t-T_\alpha)\frac{4\beta_\alpha^2}{(2-\sigma_\alpha)\chi_\alpha}})}{\frac{4\beta_\alpha^2}{(2-\sigma_\alpha)\chi_\alpha}\Pi_\gamma} \Big) \\
P_\gamma &= (1 - e^{-(\sum_{\xi=1}^{\gamma-1} \frac{2(1-\sigma_\xi)\beta_\xi}{2-\sigma_\xi}2r_\xi\Delta_\xi + \frac{(t_\gamma-T_\gamma)2r_\gamma\beta_\gamma 2(1-\sigma_\gamma)}{(2-\sigma_\gamma)})})
\end{aligned}
\tag{13}
$$

where we define:

$$
\begin{aligned}
\pi_u &= \left( \frac{r_u\beta_u 2(1-\sigma_u)}{(2-\sigma_u)} \right) \\
\Pi_\gamma &= \left( \sum_{\xi=1}^{\gamma-1} \frac{\Delta_\xi r_\xi \beta_\xi 2(1-\sigma_\xi)}{(2-\sigma_\xi)} + \frac{(t_\gamma-T_\gamma)r_\gamma\beta_\gamma 2(1-\sigma_\gamma)}{(2-\sigma_\gamma)} \right) \\
&= \left( \sum_{\xi=1}^{\gamma-1} \Delta_\xi \pi_\xi + (t_\gamma - T_\gamma)\pi_\gamma \right)
\end{aligned}
\tag{14}
$$

We can now calculate $p(\alpha|\gamma)$.

$$
\begin{aligned}
p(\alpha|\gamma) &= \int_{T_\alpha}^{T_{\alpha+1}} \frac{P_\gamma 2\beta_\alpha^2}{(2-\sigma_\alpha)\chi_\alpha} \Big( \sum_{\eta=1}^{\alpha-1} \frac{\pi_\eta(1-e^{-\Delta_\eta \frac{4\beta_\eta^2}{(2-\sigma_\eta)\chi_\eta}})(e^{\int_{T_{\eta+1}}^{t} -\frac{4\beta_v^2}{(2-\sigma_v)\chi_v}dv})}{\frac{4\beta_\eta^2}{(2-\sigma_\eta)\chi_\eta}\Pi_\gamma} \\
&\quad + \frac{\pi_\alpha(1-e^{-(t-T_\alpha)\frac{4\beta_\alpha^2}{(2-\sigma_\alpha)\chi_\alpha}})}{\frac{4\beta_\alpha^2}{(2-\sigma_\alpha)\chi_\alpha}\Pi_\gamma} \Big) dt \\
&= \frac{P_\gamma 2\beta_\alpha^2}{(2-\sigma_\alpha)\chi_\alpha} \Big( \sum_{\eta=1}^{\alpha-1} \frac{\pi_\eta(1-e^{-\Delta_\eta \frac{4\beta_\eta^2}{(2-\sigma_\eta)\chi_\eta}})(e^{\int_{T_{\eta+1}}^{T_\alpha} -\frac{4\beta_v^2}{(2-\sigma_v)\chi_v}dv})(1-e^{-\Delta_\alpha \frac{4\beta_\alpha^2}{(2-\sigma_\alpha)\chi_\alpha}})}{\frac{4\beta_\alpha^2}{(2-\sigma_\alpha)\chi_\alpha}\frac{4\beta_\eta^2}{(2-\sigma_\eta)\chi_\eta}\Pi_\gamma} \\
&\quad + \frac{\pi_\alpha(\Delta_\alpha - \frac{(1-e^{-\Delta_\alpha \frac{4\beta_\alpha^2}{(2-\sigma_\alpha)\chi_\alpha}})}{\frac{4\beta_\alpha^2}{(2-\sigma_\alpha)\chi_\alpha}})}{\frac{4\beta_\alpha^2}{(2-\sigma_\alpha)\chi_\alpha}\Pi_\gamma} \Big)
\end{aligned}
\tag{15}
$$

### Case 2: $\alpha > \gamma$

We first need $p(t|t_\gamma)$ when $\alpha > \gamma$, which is obtained as:

$$
\begin{aligned}
p(t|t_\gamma) &= \int_0^{t_\gamma} \frac{P_\gamma \pi_u \frac{2\beta_t^2}{(2-\sigma_t)\chi_t} e^{\int_u^{t_\gamma} -\frac{4\beta_v^2}{(2-\sigma_v)\chi_v}dv} e^{\int_{t_\gamma}^{t} -\frac{2\beta_v^2}{(2-\sigma_v)\chi_v}dv}}{\Pi_\gamma} du \\
&= \frac{P_\gamma \frac{2\beta_\alpha^2}{(2-\sigma_\alpha)\chi_\alpha}}{\Pi_\gamma} \Big( \sum_{\eta=0}^{\gamma-1} \int_{T_\eta}^{T_{\eta+1}} \pi_u e^{\int_u^{t_\gamma} -\frac{4\beta_v^2}{(2-\sigma_v)\chi_v}dv} e^{\int_{t_\gamma}^{t} -\frac{2\beta_v^2}{(2-\sigma_v)\chi_v}dv} du \\
&\quad + \int_{T_\gamma}^{t_\gamma} \pi_u e^{\int_u^{t_\gamma} -\frac{4\beta_v^2}{(2-\sigma_v)\chi_v}dv} e^{\int_{t_\gamma}^{t} -\frac{2\beta_v^2}{(2-\sigma_v)\chi_v}dv} du \Big) \\
&= \frac{P_\gamma \frac{2\beta_\alpha^2}{(2-\sigma_\alpha)\chi_\alpha}}{\Pi_\gamma} \Big( \sum_{\eta=0}^{\gamma-1} \int_{T_\eta}^{T_{\eta+1}} \pi_\eta e^{-(T_{\eta+1}-u)\frac{4\beta_\eta^2}{(2-\sigma_\eta)\chi_\eta}} e^{\int_{T_{\eta+1}}^{t_\gamma} -\frac{4\beta_v^2}{(2-\sigma_v)\chi_v}dv} e^{\int_{t_\gamma}^{t} -\frac{2\beta_v^2}{(2-\sigma_v)\chi_v}dv} du \\
&\quad + \int_{T_\gamma}^{t_\gamma} \pi_\gamma e^{-(t_\gamma-u)\frac{4\beta_\gamma^2}{(2-\sigma_\gamma)\chi_\gamma}} e^{\int_{t_\gamma}^{t} -\frac{2\beta_v^2}{(2-\sigma_v)\chi_v}dv} du \Big) \\
&= \frac{P_\gamma \frac{2\beta_\alpha^2}{(2-\sigma_\alpha)\chi_\alpha}}{\Pi_\gamma} \Big( \sum_{\eta=0}^{\gamma-1} \pi_\eta \frac{(1-e^{-\Delta_\eta \frac{4\beta_\eta^2}{(2-\sigma_\eta)\chi_\eta}})}{\frac{4\beta_\eta^2}{(2-\sigma_\eta)\chi_\eta}} e^{\int_{T_{\eta+1}}^{t_\gamma} -\frac{4\beta_v^2}{(2-\sigma_v)\chi_v}dv} e^{\int_{t_\gamma}^{t} -\frac{2\beta_v^2}{(2-\sigma_v)\chi_v}dv} \\
&\quad + \pi_\gamma \frac{(1-e^{-(t_\gamma-T_\gamma)\frac{4\beta_\gamma^2}{(2-\sigma_\gamma)\chi_\gamma}})}{\frac{4\beta_\gamma^2}{(2-\sigma_\gamma)\chi_\gamma}} e^{\int_{t_\gamma}^{t} -\frac{2\beta_v^2}{(2-\sigma_v)\chi_v}dv} \Big)
\end{aligned}
\tag{16}
$$

We can now calculate $p(\alpha|\gamma)$.

$$
\begin{aligned}
p(\alpha|\gamma) &= \int_{T_\alpha}^{T_{\alpha+1}} \frac{P_\gamma \frac{2\beta_\alpha^2}{(2-\sigma_\alpha)\chi_\alpha}}{\Pi_\gamma} \Big( \sum_{\eta=0}^{\gamma-1} \pi_\eta \frac{(1-e^{-\Delta_\eta \frac{4\beta_\eta^2}{(2-\sigma_\eta)\chi_\eta}})}{\frac{4\beta_\eta^2}{(2-\sigma_\eta)\chi_\eta}} e^{\int_{T_{\eta+1}}^{t_\gamma} -\frac{4\beta_v^2}{(2-\sigma_v)\chi_v}dv} e^{\int_{t_\gamma}^{t} -\frac{2\beta_v^2}{(2-\sigma_v)\chi_v}dv} \\
&\quad + \pi_\gamma \frac{(1-e^{-(t_\gamma-T_\gamma)\frac{4\beta_\gamma^2}{(2-\sigma_\gamma)\chi_\gamma}})}{\frac{4\beta_\gamma^2}{(2-\sigma_\gamma)\chi_\gamma}} e^{\int_{t_\gamma}^{t} -\frac{2\beta_v^2}{(2-\sigma_v)\chi_v}dv} \Big) dt \\
&= \frac{P_\gamma \frac{2\beta_\alpha^2}{(2-\sigma_\alpha)\chi_\alpha}}{\Pi_\gamma} \Big( \sum_{\eta=0}^{\gamma-1} \frac{\pi_\eta(1-e^{-\Delta_\eta \frac{4\beta_\eta^2}{(2-\sigma_\eta)\chi_\eta}})}{\frac{4\beta_\eta^2}{(2-\sigma_\eta)\chi_\eta}} e^{\int_{T_{\eta+1}}^{t_\gamma} -\frac{4\beta_v^2}{(2-\sigma_v)\chi_v}dv} + \frac{\pi_\gamma(1-e^{-(t_\gamma-T_\gamma)\frac{4\beta_\gamma^2}{(2-\sigma_\gamma)\chi_\gamma}})}{\frac{4\beta_\gamma^2}{(2-\sigma_\gamma)\chi_\gamma}} \\
&\quad + \int_{T_\alpha}^{T_{\alpha+1}} e^{\int_{t_\gamma}^{T_\alpha} -\frac{2\beta_v^2}{(2-\sigma_v)\chi_v}dv} e^{-(t-T_\alpha)\frac{2\beta_\alpha^2}{(2-\sigma_\alpha)\chi_\alpha}} \Big) dt \\
&= \frac{P_\gamma}{\Pi_\gamma} \Big( \sum_{\eta=0}^{\gamma-1} \frac{\pi_\eta(1-e^{-\Delta_\eta \frac{4\beta_\eta^2}{(2-\sigma_\eta)\chi_\eta}})}{\frac{4\beta_\eta^2}{(2-\sigma_\eta)\chi_\eta}} e^{\int_{T_{\eta+1}}^{t_\gamma} -\frac{4\beta_v^2}{(2-\sigma_v)\chi_v}dv} + \frac{\pi_\gamma(1-e^{-(t_\gamma-T_\gamma)\frac{4\beta_\gamma^2}{(2-\sigma_\gamma)\chi_\gamma}})}{\frac{4\beta_\gamma^2}{(2-\sigma_\gamma)\chi_\gamma}} e^{\int_{t_\gamma}^{T_\alpha} -\frac{2\beta_v^2}{(2-\sigma_v)\chi_v}dv} \Big) \\
&\quad (1 - e^{-\Delta_\alpha \frac{2\beta_\alpha^2}{(2-\sigma_\alpha)\chi_\alpha}}))
\end{aligned}
\tag{17}
$$

### Case 3: $\alpha = \gamma$

Because the sum of probabilities for the three cases sum up to one, we obtain the following formula:

$$
p(\gamma|\gamma) = 1 - \Big( \sum_{\alpha=0}^{\gamma-1} p(\alpha|\gamma) + \sum_{\alpha=\gamma+1}^{n} p(\alpha|\gamma) \Big)
\tag{18}
$$

## Emission matrix

Note as highlighted in **Sellinger et al., 2020**, that seed banking can significantly lengthen the coalescent tree. In this case the infinite site model hypothesis might be violated, and therefore we adapt the formula for the emission matrix:

$$P(0|\gamma) = e^{-2\mu(((\beta_\gamma + ((1-\beta_\gamma)\mu_b))(t_\gamma - Tc_\gamma)) + \sum_\eta^{\gamma-1}((\beta_\eta + ((1-\beta_\eta)\mu_b))\Delta_\eta))}$$

$$P(1|\gamma) = 1 - P(0|\gamma)$$

(19)

where $\mu$ is the mutation rate per nucleotide per $N$ generations, $\mu_b$ ratio of mutation rate during the dormant stage over the one in the active stage, $\beta_\gamma$ the dormancy rate in state $\gamma$, and $t\gamma$ the average coalescent time in state $\gamma$.

# Appendix 2

## ABC method

### Approximate Bayesian computation

ABC offers a statistical framework for modeling, widely used for population genetics (***Beaumont et al., 2002***). ABC addresses the two crucial modeling questions (1) which is the most supported model given the observed data, and (2) which are the most supported parameter values given the observed data from natural populations. Likelihood functions or total probabilities may be analytically complicated or computationally intractable, whereas, in the ABC framework, a posterior probability distribution is approximated using simulations. In addition, since ABC is Bayesian, it allows the implementation of prior knowledge. For this purpose, instead of using the full complexity of genetic diversity as in full-likelihood methods, the observation must be summarized into a scalar vector of summarizing statistics. The same summarizing statistics can be calculated from both simulated and natural populations. The Euclidean distance between the two simulated and observed statistics vectors determines the similarity between the simulated and true models. By defining a similarity threshold $\varepsilon$, we identify a set of simulations to explain the observed results. The distribution of parameters of those simulations approximates the posterior probability distribution. For this purpose, the ABC method could be optimized in a plethora of ways. The optimal performance of the ABC depends on the Bayesian sufficiency criterion of the summarizing statistic and the extensive exploration of the likelihood surface. Thus, we use PLS not only to reduce the dimensionality of the summary statistics vectors but also to reduce the influence of noise in the data. Moreover, we also apply multinomial regression (***Wegmann et al., 2010***) for the model choice and local linear regression for estimating the parameters. Both allow an increment of $\varepsilon$, causing an increase in the marginal posterior densities. This increase in the marginal density causes reduced suffering of the curse of dimensionality to improve approximation to the likelihood function.

### Models

We propose a transition-to-selfing model (***Figure 3A***) and a confounding population-size model (***Figure 3B***) to distinguish these two effects. The transition-to-selfing model represents a diploid, random mating population of constant size, which undergoes a transition to selfing at a given time. However, the population-size model assumes a constant selfing rate.

In contrast to A, the population-size model B undergoes one single change in population size, that is we propose two possible demographic models explaining a reduced genetic diversity after a single event in the past. In both models, A and B, we define equal prior probabilities of current population sizes; however, the effective population size in the ancestral state of the scenario of A could increase to the twofold, only, but in B, we define the identical prior probabilities of population sizes compared to the current population size. Thus, we could identify whether a transition to selfing is inferred specifically or confounded with a simple reduction in population size.

### Simulation of genetic data

If not indicated otherwise, from 20 individuals, we sampled 5 independent haplotypes of 1 Mb length mimicking five chromosomes. For the calculation of the SFS and LD we use the 20 haplotypes. The other summary statistics are based on pairwise comparison of sequences ($TL_{true}$, $TM_{true}$, $TM_{win}$). If not indicated differently we compare all possible 20 choose 2 pairs. Thus, the total pairwise length results in 950 Mb. We used mutation and recombination rates of $\mu = r = 10^{-8}$. For the ABC we created 100,000 datasets for each model. We used corresponding prior parameters for both models (***Supplementary file 1***, Table S1). We tested the performance of the ABC under the assumption of neutrality. The corresponding PODs were simulated for different transition times under the shift-to-selfing model (***Supplementary file 1***, Table S1). Thus, we obtain a time series for the performance analysis. We tested the performance for following transitioning times in generations: 1000; 2000; 3000; 4000; 5000; 6000; 7000; 8000; 9000; 10,000; 12,000; 16,000; 20,000; 30,000; 40,000; 50,000; 60,000; 70,000; 80,000; 90,000; 100,000; 200,000. That translates to coalescent time units ranging from 0.05 to 10. For each condition, we created 100 independent PODs. We tested whether the assumptions of our ABC are robust if the PODs were simulated under BGS. We used the DFEs estimated by ***Hämälä and Tiffin, 2020***, for *A. thaliana*. We simulated sets of five pseudo-genomes of 1 Mb and used the annotation file of TAIR10 to determine the spatial distribution of exons. We simulated the BGS-PODs forward-in-time under WF assumptions. We created 100 independent burn-ins of $10N$ generations. Starting from those, we simulated a time series of transitions to explicit

selfing under the same parameters as used for the neutral PODs. Each set of five simulations was aggregated and summarized into a single POD. The same set of PODS were used to test the statistical performance of *teSMC*.

## Implementation and software

Except for BGS, we simulated the genetic data using the coalescent implemented in msprime version 0.7.4 (*Kelleher et al., 2016*). We simulated the most recent 1000 generations under the discrete-time WF to avoid biases in IBD and the following times under the SMC-prime. We used the rescaled coalescent-with-selfing (*Nordborg and Donnelly, 1997*) to simulate transitions to selfing for a recent selfing past. Backward-in-time, our simulations underwent a transition to outcrossing.

For the BGS simulations, we wrote a forward-in-time simulator using *slim version 3.6* (*Haller and Messer, 2019*). We forbid accidental selfing. We used the tree-sequence-recording option. A few lineages were not coalesced after the simulation. Thus, we recapitated the obtained tree-sequences from the *pyslim-package version 0.6*, which in turn utilizes msprime.

We implemented the whole pipeline into *Snakemake 5.13*. We have run all simulations on the high-performance cluster of the MPIPZ.

## Summarization of genetic data

### SFS/LD

For each sampled set of 20 haplotypes, calculated the folded SFS and the LD. The LD was calculated as $r^2$ from a subset of 10,000 randomly chosen SNPs and discretized into discrete physical distances with following breakpoints: 6105; 11,379; 21,209; 39,531; 73,680; 137,328; 255,958; 477,066; 889,175 bp. Because recombination rates are not constant through time, we cannot relate physical distance to coalescent times. Thus, we set the discretization boundaries to fixed values.

### TM$_{win}$

For each pairwise comparison, we counted the number of SNPs for a sliding non-overlapping window of 10,000 bp. The ABC depends on vectors of scalar summary statistics of finite size. Thus, we discretized the diversity into $m$ segments, each reflecting the expected diversity under a given coalescent time. The discretization was derived from the quantiles of the exponential distribution in order to obtain an equal information distribution per discretized diversity window via $-f * log_e \frac{1-i}{m}$. If $f = 1$, the diversity scales in coalescent times with mutation rate 1. To scale the discretization breakpoints we used $f = 8 \cdot L_{WIN} \cdot N \cdot \mu$ with $\mu$, $N$, $L_{WIN}$ being the per generation per bp mutation rate, the population size, and the window size, respectively. We usually choose $f$ to provide Bayesian sufficiency within the time frame of expected estimates of times of transition to selfing. For the performance analysis of the ABC, we chose $N = 40,000$. Further, $i$ and $m$ were defined as $i \in \{1 \ldots m\}$ and $m = 20$ bins. We only kept discretization breakpoints with at least a single integer value in-between neighboring breakpoints. Further, we obtained the transition proportion of the $i$th to the $(i+1)$ th window for each of the $m$ discrete diversity bins. Thus, we obtained a $m^2$ transition matrix. We flattened the matrix into a one-dimensional vector of maximal length $m^2$.

## Dimensionality reduction

The dimensionality of our summary statistics TL-distribution, TM$_{true}$, and TM$_{win}$ is up to 400 for each. To overcome the curse of dimensionality in our ABC, first, we centralized, normalized, and Box Cox transformed each calculated statistic to obtain the orthogonal independent variation. Then, we applied the PLS analysis to our data as suggested from *Wegmann et al., 2010*. According to the chosen combination of summary statistics, we chose an appropriate number of model selection and parameter estimation components. SFS and LD are summarizations of lower dimension. Thus, we did not necessarily apply a dimensionality reduction on them.

## Model choice

The Bayesian framework naturally allows obtaining a posterior density for each proposed model. The Bayes factor is simply defined as the ratio of the marginal densities of two models. In ABC, we approximate those marginal densities by the posterior density. Thus, the Bayes factor $B_{AB} = \frac{dens_A \left( stats_{obs} \right)}{dens_B \left( stats_{obs} \right)}$ provides an approximation to the model support given the observed data

(compare *Wegmann et al., 2010*). We categorize the Bayes factors into negative, barley worth mentioning, substantial, strong, very strong, and decisive (*Jeffreys, 1998*).

We calculated the Bayes factor for different sets of summarizing statistics, performed a model choice using a multinomial regression analysis between proposed models A and B. The calculations were done using the R package abc (*Csilléry et al., 2012*). We accepted 1% of the total number of simulations for the model choice.

## Parameter inference

We tested the performance of parameter inference under model 1 (*Figure 3*) described in the model selection section (Statistical methods to estimate the age of a transition to selfing: tsABC) to estimate the date of a transition to selfing. We conducted the estimation using the R package abc (*Csilléry et al., 2010*). We accepted the closest 1% of the simulations of the transitioning model. For each corresponding set of summarizing statistics, we estimated the average posterior distribution for the 100 PODs. We show the average quantiles for the following credible intervals: 99%, 95%, 90%, 80%, 50%, 25%, 10% and the median for the whole time series. We show the performance for each parameter of the model using following sets of summary statistics: SFS/LD, $TM_{win}$, and both combined. For each we use an individual set of PLS components: no dimensionality reduction for SFS/LD, 20-PLS for $TM_{win}$, and the combined set.

## Performance/accuracy analysis

Transitions to selfing result in a reduction of diversity. To test whether our ABC approach identifies transitions to selfing against a model of population census reduction, we conduct a model choice experiment. We approximate and compare the posterior densities of the transition to selfing model with constant population size (*Figure 3C–E*) to a model which undergoes a change in population size but is constant in selfing (*Figure 3*). We calculated the Bayes factors using multinomial logistic regression. The results (*Figure 3*) depict the proportion of the correct model estimations. Depending on our summarizing statistics, our results indicate that we can precisely detect transitions to selfing for times up to $2.5N_e$ generations in the past if $r/\mu=1$.

Interestingly, when using SFS/LD, we precisely detect the correct model for very recent times and outperform any other combination of parameters. However, $TM_{win}$ enables us to detect also older transitions to selfing. With the combination of both summarizations, we maintain a consistently high performance to model selection. The power and specificity of our model selection are more uncertain if the ratio $r/\mu$ increases.

After determining the correct model, we estimated the actual parameter of that model: When did a population undergo a transition to selfing? To test the performance of estimating the parameters, we reused the previously simulated PODs and calculated several credibility intervals for each different time point.

