## [Editor Report]

This manuscript details an important development of population genetics theory that can be used to infer past changes in the selfing rate of natural populations. The inference procedure is convincing and represents a substantial improvement upon previous methods. The work will be of broad interest to researchers studying mating system evolution and its consequences and will improve demographic inferences drawn from population genetic approaches.

---

## [Decision Letter]

**Decision letter after peer review:**

Thank you for submitting your article "Inference of evolutionary transitions to self-fertilization using whole-genome sequences" for consideration by *eLife*. We apologize for the uncharacteristic delay in the handling of your manuscript, which was due to the absence of an editor. Your article has now been reviewed by two peer reviewers, and the evaluation has been overseen by a Reviewing Editor and Molly Przeworski as the Senior Editor. The following individual involved in the review of your submission has agreed to reveal their identity: Takashi Tsuchimatsu (Reviewer #1).

Essential revisions:

Both reviewers appreciated an important contribution to the study of mating systems transitions, and the welcome addition of promising new methods to detect ancestral changes in the rate of selfing. In spite of these promises, however, they also judged that the current manuscript did not reach its full potential and needed major revisions on two main aspects:

1. The comparison with the estimate of the time since *A. thaliana* has been a selfer by Bechsgaard et al. (2006) needs to be more careful and comprehensive. This includes in particular the lack of a proper test beyond the comparison of point estimates, and the use of different mutation rates. The importance of this latter point was already noted in a previous review (Shimizu and Tsuchimatsu 2015; Ann Rev Eco Evo Syst), which is missing from the current version. They also called for an evaluation of the consequences of the (arbitrary) choice of a limited number of genomic regions, and a more formal comparison of the ancestral population size estimated by their method with those available from the literature for that species.

2. Reviewer 3 noted a number of missing references that are required to better anchor the present developments in the previous theoretical literature, and made a series of precise suggestions with that regard. Along this line, the manuscript should better highlight that the « core » process upon which the estimation procedure is based (the effect of selfing on the association between segment length and TMRCA) is currently evaluated mostly through computational approximations. The theoretical treatment of this relationship is currently relegated to an appendix, the presentation of which could be improved greatly.

*Reviewer #1 (Recommendations for the authors):*

1) L65-71: I would suggest citing Stebbins' work because he is the pioneer who studied the frequent transition to selfing and the evolutionary dead-end.

2) Always italicize S of "S-locus".

3) L93: This would be 413,000 years instead of 430,000 years.

4) It is an important finding that population size is more accurately inferred when the outcrossing-to-selfing transition is considered. As discussed, this would be potentially important for other organisms including crops, as the domestication process often involves a shift in mating systems. In that sense, authors could discuss how the estimated population size change of *Arabidopsis thaliana* (Figure 5B) is (or is not) consistent with previous estimates of population size.

*Reviewer #2 (Recommendations for the authors):*

As the authors did not even provide page numbers, I use line numbers to indicate the start of sections.

With regard to Theme (1), equations [1] and [2] are attributed to Nordborg, but Fu (1997 Genetics) and especially Golding and Strobeck (1980 Genetics) should also be cited.

The authors use [1] and [2] to develop a sequential Markov coalescent algorithm for estimating ancestral population size and the time since a change in the level of selfing.

Once again, the writing suggests Theme (1). In the Results section (L 202), a rain of references to sequential Markov methods appears, with little discussion of their relationship to the authors' work. Such scholarly discussion might be expected to appear in the Introduction or Discussion sections, but do not. For example, Palacios et al. (2015 Genetics) are relevant to the establishment of a solid inferential framework for approaches of this kind.

The authors address teSMC in a manner that appears to relegate it to mainly theoretical interest because the age of the MRCA of a chromosomal segment (TMRCA) is not in general known. Even so, the authors detect a possibly significant trend in the course of forward simulations of the process: a change in selfing rate induces a change in the magnitude of the negative association between segment length and TMRCA. They seek to use this phenomenon as a signature of a shift in the mating system apart from a shift in population size.

With regard to Theme (2), the paper would have been improved if they had undertaken to explore the basis of this relationship theoretically. The paper only provides an opaque description:

"the probability of a recombination event is not increasing linearly with time" (L 178).

The rate of recombination (r) is in fact not changing at all, and "time" here may (or may not) mean TMRCA. The ambiguity of this sentence leaves considerable room for guessing, and the reader should not have to guess at all. It is ρ_\σ (which is proportional to the probability that the next evolutionary event back in time is a recombination event rather than a coalescence event) that is affected by selfing. Both ρ_\σ and TMRCA decline as selfing increases, but the change in selfing rate has less effect on TMRCA. Is this what L 178 means? In any case, it should surely be stated more clearly.

A few lines later, we have

"We also made the important observation that all the segments that coalesce in the outcrossing phase, trace back their ancestry to a subset of segments that do not coalesce more recently

than t_\σ…." (L 188)

Are the authors saying that segments that coalesce before t_\σ do not coalesce after t_\σ? These are just a few examples (on the same page) of what are NOT minor grammatical lapses but major barriers to communication. They give the impression of sloppiness or indifference to the reader.

The authors note (L 275)

"Unfortunately, while the lengths of TMRCA-segments are straightforward to calculate on simulated genealogies (Figures 1A, B), it is more difficult to estimate them based on genomic diversity data alone."

Unfortunate or not, observing genetic diversity and not TMRCAs is of course the relevant case. While statistical uncertainty does not appear to be addressed for teSMC, it is addressed in the authors' ABC implementation of their approach (tsABC). As this section does deal with basing inferences on observations of genetic diversity (rather than TMRCA), it has greater relevance to the analysis of real data.

Even so, the description (L 288) in the main text suggests only that results were good, but not why or even exactly what the results were. In Figure 3, the reader must glean from the caption exactly what was done and what is being shown. Exactly what is on the X-axis is unclear. The t_\σ at the far right might suggest that the X-axis represents the true time since the switch to selfing, but it isn't clear whether the units are in generations or years or some multiple of those units. Figure 3 C/D/E appears to depict support for the true t_\σ, but not support for incorrect values. Figure 3 F/G/H does speak to whether t_\σ can be inferred, but the posterior ranges shown seem to be quite wide. After all this eyeballing of the Figure, a reader might be less entirely convinced that the results are as supportive of tsABC as the text suggests.

The next section (L 316) goes off on a tangent regarding background selection. This aspect, while important, might be reserved for a separate study: one in which a rigorous exploration might be conducted. As presented, it is not entirely clear what was done. It appears that genomic data were simulated under a background selection model. What the authors refer to as "robustness" appears to correspond to obtaining similar results from teSMC or tsABC using masked or unmasked data. The term "masking" seems to suggest that only sites NOT under selection were given to teSMC or tsABC, even though those sites were subject to background selection. If this is correct, then the finding that masking versus unmasking gives about the same results does not address the question of model misspecification: both masked and unmasked data sets could give equally bad inferences.

It seems that a more appropriate test would involve simulating data with and without BGS. The question is then whether the masked BGS data give similar results to the full non-BGS data. It is difficult to ascertain whether or not this is what the authors did.

The description of the analysis of real data is confined to a single paragraph (L 349). That the method suggested that self-compatibility arose in the *Arabidopsis thaliana* lineage sometime between previous estimates (413 KYA and 1000 KYA) seems a rather low bar. The authors claim "remarkable" (L 413) agreement between their results and the 413 KYA figure, which was obtained from a model-based analysis of variation at the *S-locus*. However, the authors do not even give credible intervals for their estimates. Giving the authors the benefit of the doubt, one might guess that the X-axis in Figure 5 is in units of 10^4 years and that the 95% credible interval is perhaps (55 KYA, 60 KYA).

Does this mean that the Bechsgaard estimate of 41 KYA lies outside the authors' credible range? Note that the Bechsgaard figure assumes the rough estimate of 5 MY for the divergence of *A. thaliana* from its SI relatives, so that a slight revision of this figure could bring their estimate closer to (or farther from) those of the authors. That no comment or discussion about the results of real data appears bolsters the impression of Themes (1) and (3).

The Discussion begins around L 367. Once again, I am supportive of this submission as primarily theoretical, with the very short treatment of the Arabidopsis data intended only as a worked example. However, the authors seem to have (much) greater ambitions. If the authors actually wish to characterize their work as a breakthrough that opens up long-standing questions to rigorous analysis, then they need to address in detail inferences made on the basis of real data.

For the Arabidopsis analysis (the only real data application), do the authors regard their estimate as superior to the model-based estimate of Bechsgaard? How seriously should the estimates of the changes in population size (Figure 5B) be taken? If the authors wish to argue that the changes in population size are real, they might at least provide credible intervals for the estimates and explore what is known about the ecological history of Arabidopsis.

Authors concede (L 312) that their ABC implementation could not infer the ancestral selfing rate. In the Arabidopsis case, this rate is assumed to be zero, as appropriate for a functional self-incompatibility (SI) system. Does this mean, then, that the method is useful mainly for cases in which the ancestral selfing rate is somehow known? Is their method actually only applicable to Arabidopsis?

Rather than addressing such points, the Discussion appears to cast their method permitting access to a very wide range of questions, including the evolution of sex (Barton and Charlesworth 1998). Exactly what the basis for this claim is unclear, especially since the breeding system shift modeled is wholly sexual (meiotic).

Perhaps the most germane suggestion is for the authors to determine whether this submission is primarily whatever the Discussion is maintaining or a theoretical exploration with a bit of illustration using real data.

If the latter (my view), then the theory sections might be improved. They presently appear in the Appendices as dumps of notes, with little effort invested in concise exposition. With regards to Theme (2), when presented with an opportunity for theoretical exploration, the authors appear to choose to rely on computation: the lack of theoretical exploration of the trends noted in Figure 1 and resorting to ABC over a model-based approach come to mind. I suggest that taking a different tack would facilitate deeper insight.

---

## [Author Response]

Essential revisions:Both reviewers appreciated an important contribution to the study of mating systems transitions, and the welcome addition of promising new methods to detect ancestral changes in the rate of selfing. In spite of these promises, however, they also judged that the current manuscript did not reach its full potential and needed major revisions on two main aspects:1. The comparison with the estimate of the time since *A. thaliana* has been a selfer by Bechsgaard et al. (2006) needs to be more careful and comprehensive. This includes in particular the lack of a proper test beyond the comparison of point estimates, and the use of different mutation rates. The importance of this latter point was already noted in a previous review (Shimizu and Tsuchimatsu 2015; Ann Rev Eco Evo Syst), which is missing from the current version. They also called for an evaluation of the consequences of the (arbitrary) choice of a limited number of genomic regions, and a more formal comparison of the ancestral population size estimated by their method with those available from the literature for that species.

We thank the editor and reviewers to acknowledge the relevance of the manuscript topic as well as for their constructive criticism of our work. We have a addressed this issues in the revised manuscript (see our answers below)

2. Reviewer 3 noted a number of missing references that are required to better anchor the present developments in the previous theoretical literature, and made a series of precise suggestions with that regard. Along this line, the manuscript should better highlight that the « core » process upon which the estimation procedure is based (the effect of selfing on the association between segment length and TMRCA) is currently evaluated mostly through computational approximations. The theoretical treatment of this relationship is currently relegated to an appendix, the presentation of which could be improved greatly.

We thank reviewer 2 for the sharp comments, which improved the focus of the manuscript. We now increased the first result section by describing the theoretical results and their implication in more depth (was before in the Appendix) on page 7. We also improve the literature part by citing ad hoc references. We also moved (as suggested by reviewer 2) some literature part on SMC and ABC to the introduction (before in the result section) to streamline our result part.

Reviewer #1 (Recommendations for the authors):1) L65-71: I would suggest citing Stebbins' work because he is the pioneer who studied the frequent transition to selfing and the evolutionary dead-end.

Done on p. 3 l.6,

2) Always italicize S of "S-locus".

Corrected.

3) L93: This would be 413,000 years instead of 430,000 years.

Corrected at p.4 l.9.

4) It is an important finding that population size is more accurately inferred when the outcrossing-to-selfing transition is considered. As discussed, this would be potentially important for other organisms including crops, as the domestication process often involves a shift in mating systems. In that sense, authors could discuss how the estimated population size change of *Arabidopsis thaliana* (Figure 5B) is (or is not) consistent with previous estimates of population size.

We now compare our estimation of effective population sizes for *A. thaliana* with previously published estimates. P. 13 l. 16-31

Reviewer #2 (Recommendations for the authors):As the authors did not even provide page numbers, I use line numbers to indicate the start of sections.

We now include page and line numbers.

With regard to Theme (1), equations [1] and [2] are attributed to Nordborg, but Fu (1997 Genetics) and especially Golding and Strobeck (1980 Genetics) should also be cited.

Citations were added at p. 4 l. 31 – p. 5 l. 4

The authors use [1] and [2] to develop a sequential Markov coalescent algorithm for estimating ancestral population size and the time since a change in the level of selfing.Once again, the writing suggests Theme (1). In the Results section (L 202), a rain of references to sequential Markov methods appears, with little discussion of their relationship to the authors' work. Such scholarly discussion might be expected to appear in the Introduction or Discussion sections, but do not. For example, Palacios et al. (2015 Genetics) are relevant to the establishment of a solid inferential framework for approaches of this kind.

Reply: We cite now the key references by Palacios et al. (2015) and Gattepaille et al. (2016)(p. 6 l. 10, p. 6 l. 5), which provide indeed a solid foundation for the SMC and interpretation of TMRCA. We have improved the introduction to explicit the difference between our method and those from the literature. (p. 5 l. 34 – p. 6 l. 10). We had discussed the link and convergence of results between our approach and the work by Gattepaille et al. in a previous article (Sellinger et al. Mol Ecol Res 2020).

The authors address teSMC in a manner that appears to relegate it to mainly theoretical interest because the age of the MRCA of a chromosomal segment (TMRCA) is not in general known. Even so, the authors detect a possibly significant trend in the course of forward simulations of the process: a change in selfing rate induces a change in the magnitude of the negative association between segment length and TMRCA. They seek to use this phenomenon as a signature of a shift in the mating system apart from a shift in population size.

We thank the reviewer for this summary of our work and the suggestion to explore in more depth the theoretical basis for our inference. We now enhance the theoretical content of the first part of the results. We start now from the theory and then provide the results from simulations, see lines p. 7 l. 6- p.8 l. 33

With regard to Theme (2), the paper would have been improved if they had undertaken to explore the basis of this relationship theoretically. The paper only provides an opaque description:"the probability of a recombination event is not increasing linearly with time" (L 178).

We thank the reviewer for this suggestion. We now rephrased the section to clearly state our new theoretical results (p. 7 l. 6- p.8 l. 33) and updated Figure 1—figure supplement 1 for visual support. The transitions in Figure 1—figure supplement 1 are now computed from the equation 3 in the main text.

The rate of recombination (r) is in fact not changing at all, and "time" here may (or may not) mean TMRCA. The ambiguity of this sentence leaves considerable room for guessing, and the reader should not have to guess at all. It is ρ_\σ (which is proportional to the probability that the next evolutionary event back in time is a recombination event rather than a coalescence event) that is affected by selfing. Both ρ_\σ and TMRCA decline as selfing increases, but the change in selfing rate has less effect on TMRCA. Is this what L 178 means? In any case, it should surely be stated more clearly.

We apologize for the confusion and changed vocabulary to be more rigorous. We admit that we did not find previously the right balance between in depth precise mathematical explanations and an enhanced readability of our paper for a biological audience (as required for a journal as *eLife*). What we meant was the rate at which a break in the genealogy occurs. This section and others have been rewritten to be precise in our notations (p. 7 l. 6- p.8 l. 33). The first part of the results is now dedicated to this aspect with the introduction of equation 3.

A few lines later, we have"We also made the important observation that all the segments that coalesce in the outcrossing phase, trace back their ancestry to a subset of segments that do not coalesce more recentlythan t_\σ…." (L 188)Are the authors saying that segments that coalesce before t_\σ do not coalesce after t_\σ? These are just a few examples (on the same page) of what are NOT minor grammatical lapses but major barriers to communication. They give the impression of sloppiness or indifference to the reader.

We have rewritten few sentences to add precision. P. 8 l. 1-14

The authors note (L 275)"Unfortunately, while the lengths of TMRCA-segments are straightforward to calculate on simulated genealogies (Figures 1A, B), it is more difficult to estimate them based on genomic diversity data alone."Unfortunate or not, observing genetic diversity and not TMRCAs is of course the relevant case. While statistical uncertainty does not appear to be addressed for teSMC, it is addressed in the authors' ABC implementation of their approach (tsABC). As this section does deal with basing inferences on observations of genetic diversity (rather than TMRCA), it has greater relevance to the analysis of real data.

It is true that the uncertainty of TMRCA inference is not seen in SMC methods, as they rely on multiple observations to estimate the time of the hidden states (and thus provide by likelihood estimation one value and not a confidence interval). The uncertainty is addressed in our case by running several analyses on different datasets with either the same simulation parameters or on several regions of the *A. thaliana* genome as statistical replicates. The reviewer is correct that this is a key point of our study and why we provide both complementary approaches here.

Even so, the description (L 288) in the main text suggests only that results were good, but not why or even exactly what the results were. In Figure 3, the reader must glean from the caption exactly what was done and what is being shown. Exactly what is on the X-axis is unclear. The t_\σ at the far right might suggest that the X-axis represents the true time since the switch to selfing, but it isn't clear whether the units are in generations or years or some multiple of those units. Figure 3 C/D/E appears to depict support for the true t_\σ, but not support for incorrect values. Figure 3 F/G/H does speak to whether t_\σ can be inferred, but the posterior ranges shown seem to be quite wide. After all this eyeballing of the Figure, a reader might be less entirely convinced that the results are as supportive of tsABC as the text suggests.

We have updated Figure 3 and its legend to clarify the meaning of the axes and the units. Figure 3 C/D/E show the statistical support for the right model (i.e. transition to selfing) when compared to a stepwise change in N in an ABC model choice procedure. Figure 3 F/G/H shows average quantiles for 100 posteriors on t_\σ and panel H indeed indicates the precision that can be expected for t_σ. We note that the average mode of the 100 posteriors tracks the true value (solid black line in panel H) over the whole range of t_\σ values, that a large proportion of the probability density is centered around the true value (i.e. the posterios are peaky), and that the precision is improved when using TM_win verus a combination of SFS and LD.

The next section (L 316) goes off on a tangent regarding background selection. This aspect, while important, might be reserved for a separate study: one in which a rigorous exploration might be conducted. As presented, it is not entirely clear what was done. It appears that genomic data were simulated under a background selection model. What the authors refer to as "robustness" appears to correspond to obtaining similar results from teSMC or tsABC using masked or unmasked data. The term "masking" seems to suggest that only sites NOT under selection were given to teSMC or tsABC, even though those sites were subject to background selection. If this is correct, then the finding that masking versus unmasking gives about the same results does not address the question of model misspecification: both masked and unmasked data sets could give equally bad inferences.It seems that a more appropriate test would involve simulating data with and without BGS. The question is then whether the masked BGS data give similar results to the full non-BGS data. It is difficult to ascertain whether or not this is what the authors did.

The simulations used in Figure 4 have indeed been done with BGS. Details about this analysis can be found in the legend of Figure 4, in the main text (methods p. 18 l. 9-17), and in Appendix 2 (section: simulation of genetic data). What we refer to as robustness does not relate to the comparison between the masked and unmasked data, but rather, as you suggest, to the comparison of the performance of both methods when applied to data simulated with and without BGS. The case without BGS can be seen in Figure 2A for teSMC and figure 3H for tsABC. We have updated the main text to make this clearer (p. 12 l. 3-24).

The description of the analysis of real data is confined to a single paragraph (L 349).That the method suggested that self-compatibility arose in the *Arabidopsis thaliana* lineage sometime between previous estimates (413 KYA and 1000 KYA) seems a rather low bar. The authors claim "remarkable" (L 413) agreement between their results and the 413 KYA figure, which was obtained from a model-based analysis of variation at the S-locus. However, the authors do not even give credible intervals for their estimates. Giving the authors the benefit of the doubt, one might guess that the X-axis in Figure 5 is in units of 10^4 years and that the 95% credible interval is perhaps (55 KYA, 60 KYA).

This part has been revised following criticism from reviewer 1 (see answers to Reviewer 1, and the section “Application to *A. thaliana*” (p. 12 l. 26)). Credibility intervals for t_\σ are given in table 1. Labelling of figure 5 has been updated.

Does this mean that the Bechsgaard estimate of 41 KYA lies outside the authors' credible range? Note that the Bechsgaard figure assumes the rough estimate of 5 MY for the divergence of *A. thaliana* from its SI relatives, so that a slight revision of this figure could bring their estimate closer to (or farther from) those of the authors. That no comment or discussion about the results of real data appears bolsters the impression of Themes (1) and (3).The Discussion begins around L 367. Once again, I am supportive of this submission as primarily theoretical, with the very short treatment of the Arabidopsis data intended only as a worked example. However, the authors seem to have (much) greater ambitions. If the authors actually wish to characterize their work as a breakthrough that opens up long-standing questions to rigorous analysis, then they need to address in detail inferences made on the basis of real data.

Bechsgaard indeed used a different mutation rate than the one assumed in our study. This issue was raised by reviewer 1 as well and we have addressed this in the revised version. (p. 12 l. 26)

For the Arabidopsis analysis (the only real data application), do the authors regard their estimate as superior to the model-based estimate of Bechsgaard? How seriously should the estimates of the changes in population size (Figure 5B) be taken? If the authors wish to argue that the changes in population size are real, they might at least provide credible intervals for the estimates and explore what is known about the ecological history of Arabidopsis.

This issue was raised by reviewer 1 as well and was considered in the revised version (see answer to reviewer 1 and p. 13 l. 16-31 of the main text).

Authors concede (L 312) that their ABC implementation could not infer the ancestral selfing rate. In the Arabidopsis case, this rate is assumed to be zero, as appropriate for a functional self-incompatibility (SI) system. Does this mean, then, that the method is useful mainly for cases in which the ancestral selfing rate is somehow known? Is their method actually only applicable to Arabidopsis?

The method is well-suited for cases in which the transition occurs owing to a breakdown of a self-incompatibility mechanism. In such cases the selfing rate before the breakdown should be close to zero because of the self-incompatibility mechanism. Breakdown of SI is very common beyond Brassicaceae, as well documented for example in Solanaceae (Igic and Kohn 2006, Goldberg et al. 2010) or Asteraceae (Ferrer and Good-Avila 2006) So, the method is a priori suitable for many species.

Rather than addressing such points, the Discussion appears to cast their method permitting access to a very wide range of questions, including the evolution of sex (Barton and Charlesworth 1998). Exactly what the basis for this claim is unclear, especially since the breeding system shift modeled is wholly sexual (meiotic).Perhaps the most germane suggestion is for the authors to determine whether this submission is primarily whatever the Discussion is maintaining or a theoretical exploration with a bit of illustration using real data.If the latter (my view), then the theory sections might be improved. They presently appear in the Appendices as dumps of notes, with little effort invested in concise exposition. With regards to Theme (2), when presented with an opportunity for theoretical exploration, the authors appear to choose to rely on computation: the lack of theoretical exploration of the trends noted in Figure 1 and resorting to ABC over a model-based approach come to mind. I suggest that taking a different tack would facilitate deeper insight.

Thank you for the suggestion. We hope our changes and addition of a result section with theoretical results has improved our message. We also worked on the Appendix to provide better explanation for the formulae.